# Discrete-state Continuous-time Diffusion for Graph Generation

**Zhe Xu**[*]    **Ruizhong Qiu**[*]    **Yuzhong Chen**[†]    **Huiyuan Chen**[†]    **Xiran Fan**[†]    **Menghai Pan**[†]

**Zhichen Zeng**[*]              **Mahashweta Das**[†]              **Hanghang Tong**[*]

## Abstract

Graph is a prevalent discrete data structure, whose generation has wide applications such as drug discovery and circuit design. Diffusion generative models, as an emerging research focus, have been applied to graph generation tasks. Overall, according to the space of *states* and *time* steps, diffusion generative models can be categorized into discrete-/continuous-state discrete-/continuous-time fashions. In this paper, we formulate the graph diffusion generation in a discrete-state continuous-time setting, which has never been studied in previous graph diffusion models. The rationale of such a formulation is to preserve the discrete nature of graph-structured data and meanwhile provide flexible sampling trade-offs between sample quality and efficiency. Analysis shows that our training objective is closely related to the generation quality and our proposed generation framework enjoys ideal invariant/equivariant properties concerning the permutation of node ordering. Our proposed model shows competitive empirical performance against state-of-the-art graph generation solutions on various benchmarks and at the same time can flexibly trade off the generation quality and efficiency in the sampling phase.

## 1 Introduction

Graph generation has been studied for a long time with broad applications, based on either the one-shot (i.e., one-step) [50, 39, 56, 51, 72, 32] or auto-regressive generation paradigm [82, 29, 42, 52]. The former generates all the graph components at once and the latter does that sequentially. A recent trend of applying diffusion generative models [67, 23, 70] to graph generation tasks attracts increasing attentions because of its excellent performance and solid theoretical foundation. In this paper, we follow the one-shot generation paradigm, the same as most graph diffusion generative models.

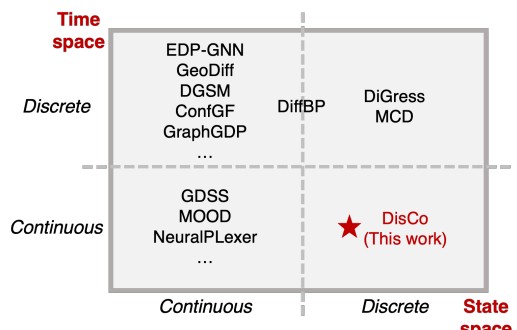

Figure 1: A taxonomy of graph diffusion models.

Some earlier attempts at graph diffusion models treat the graph data in a continuous state space by viewing the graph topology and features as continuous variables [56]. Such a formulation departs from the discrete nature of graph-structured data; e.g., topological sparsity is lost and the discretization in the generation process requires extra hyper-parameters. DiGress [73] is one of the early efforts

---

[*]University of Illinois Urbana-Champaign. {zhexu3, rq5, zhichenz, htong}@illinois.edu

[†]Visa Research. {yuzchen, hchen, xirafan, menpan, mahdas}@visa.com

38th Conference on Neural Information Processing Systems (NeurIPS 2024).

applying discrete-state diffusion models to graph generation tasks and is the current state-of-the-art graph diffusion generative model. However, DiGress is defined in the discrete time space whose generation is inflexible. This is because, its number of sampling steps must match the number of forward diffusion steps, which is a fixed hyperparameter after the model finishes training. A unique advantage of the continuous-time diffusion models [70, 32] lies in their flexible sampling process, and its simulation complexity is proportional to the number of sampling steps, determined by the step size of various numerical approaches (e.g., $\tau$-leaping [18, 8, 71]) and decoupled from the models' training. Thus, a discrete-state continuous-time diffusion model is highly desirable for graph generation tasks.

Driven by the recent advance of continuous-time Markov Chain (CTMC)-based diffusion generative model [8], we incorporate the ideas of CTMC into the corruption and denoising of graph data and propose the first discrete-state continuous-time graph diffusion generative model. It shares the same advantages as DiGress by preserving the discrete nature of graph data and meanwhile overcomes the drawback of the nonadjustable sampling process in DiGress. This Discrete-state Continuous-time graph diffusion model is named DISCO.

DISCO bears several desirable properties and advantages. First, despite its simplicity, the training objective has a rigorously proved connection to the sampling error. Second, its formulation includes a parametric graph-to-graph mapping, named backbone model, whose input-output architecture is shared between DISCO and DiGress. Therefore, the graph transformer (GT)-based backbone model [54] from DiGress can be seamlessly plugged into DISCO. Third, a concise message-passing neural network backbone model is explored with DISCO, which is simpler than the GT backbone and has decent empirical performance. Last but not least, our analyses show that the forward and reverse diffusion process in DISCO can retain the permutation-equivariant/invariant properties for its training loss and sampling distribution, both of which are critical and practical inductive biases on graph data.

Comprehensive experiments on plain and molecule graphs show that DISCO can obtain competitive or superior performance against state-of-the-art graph generative models and provide additional sampling flexibility. Our main contributions are summarized:

- **Model.** We propose the first discrete-state continuous-time graph diffusion model, DISCO. We utilize the successful graph-to-graph neural network architecture from DiGress and further explore a new lightweight backbone model with decent efficacy.
- **Analysis.** Our analysis reveals (1) the key connection between the training loss and the approximation error (Theorem 3.3) and (2) invariant/equivariant properties of DISCO in terms of the permutation of nodes (Theorems 3.8 and 3.9).
- **Experiment.** Extensive experiments validate the empirical performance of DISCO.

## 2 Preliminaries

### 2.1 Discrete-State Continuous-time Diffusion Models

A $D$-dimensional discrete state space is represented as $\mathcal{X} = \{1, \ldots, C\}^D$. A continuous-time Markov Chain (CTMC) $\{\mathbf{x}_t = [x_t^1, \cdots x_t^D]\}_{t \in [0,T]}$ is characterized by its (time-dependent) rate matrix $\mathbf{R}_t \in \mathbb{R}^{|\mathcal{X}| \times |\mathcal{X}|}$. Here $\mathbf{x}_t$ is the state at the time step $t$. The transition probability $q_{t|s}$ between from time $s$ to $t$ satisfies the Kolmogorov forward equation, for $s < t$,

$$\frac{d}{dt} q_{t|s}(\mathbf{x}_t | \mathbf{x}_s) = \sum_{\xi \in \mathcal{X}} q_{t|s}(\xi | \mathbf{x}_s) \mathbf{R}_t(\xi, \mathbf{x}_t), \tag{1}$$

The marginal distribution can be represented as $q_t(\mathbf{x}_t) = \sum_{\mathbf{x}_0 \in \mathcal{X}} q_{t|0}(\mathbf{x}_t | \mathbf{x}_0) \pi_{\texttt{data}}(\mathbf{x}_0)$ where $\pi_{\texttt{data}}(\mathbf{x}_0)$ is the data distribution. If the CTMC is defined in time interval $[0, T]$ and if the rate matrix $\mathbf{R}_t$ is well-designed, the final distribution $q_T(\mathbf{x}_T)$ can be close to a tractable reference distribution $\pi_{\texttt{ref}}(\mathbf{x}_T)$, e.g., uniform distribution. We notate the reverse stochastic process as $\tilde{\mathbf{x}}_t = \mathbf{x}_{T-t}$; a well-known fact (e.g., Section 5.9 in [63]) is that the reverse process $\{\tilde{\mathbf{x}}_t\}_{t \in [0,T]}$ is also a CTMC, characterized by the reverse rate matrix: $\tilde{\mathbf{R}}_t(\mathbf{x}, \mathbf{y}) = \frac{q(\mathbf{y})}{q(\mathbf{x})} \mathbf{R}_t(\mathbf{y}, \mathbf{x})$. The goal of the CTMC-based diffusion models is an accurate estimation of the reverse rate matrix $\tilde{\mathbf{R}}_t$ so that new data can be generated by sampling the reference distribution $\pi_{\texttt{ref}}$ and then simulating the reverse CTMC [16, 17, 18, 1]. However, the complexity of the rate matrix is prohibitively high because there are $C^D$ possible states. A reasonable simplification is to factorize the process over dimensions [8, 71, 73, 2]. Specifically,

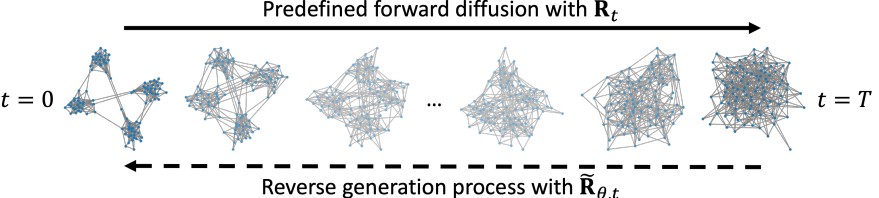

Predefined forward diffusion with $\mathbf{R}_t$

Reverse generation process with $\tilde{\mathbf{R}}_{\theta,t}$

$t = 0$          ...          $t = T$

Figure 2: An overview of DISCO. A transition can happen at any time in $[0, T]$.

the forward process is factorized as $q_{t|s}(\mathbf{x}_t|\mathbf{x}_s) = \prod_{d=1}^{D} q_{t|s}(x_t^d|x_s^d)$, for $s < t$. Then, the forward diffusion of each dimension is independent and is governed by dimension-specific forward rate matrices $\{\mathbf{R}_t^d\}_{d=1}^{D}$. With such a factorization, the goal is to estimate the dimension-specific reverse rate matrices $\{\tilde{\mathbf{R}}_t^d\}_{d=1}^{D}$.

The dimension-specific reverse rate is represented as $\tilde{\mathbf{R}}_t^d(x^d, y^d) = \sum_{x_0^d} \mathbf{R}_t^d(y^d, x^d) \frac{q_{t|0}(y^d|x_0^d)}{q_{t|0}(x^d|x_0^d)} q_{0|t}(x_0^d|\mathbf{x})$. Campbell et al. [8] estimate $q_{0|t}(x_0^d|\mathbf{x})$ via a neural network $p_\theta$ such that $p_\theta(x_0^d|\mathbf{x}, t) \approx q_{0|t}(x_0^d|\mathbf{x})$; Sun et al. [71] propose another singleton conditional distribution-based objective $\frac{p_\theta(y^d|\mathbf{x}^{\backslash d}, t)}{p_\theta(x^d|\mathbf{x}^{\backslash d}, t)} \approx \frac{q(y^d|\mathbf{x}^{\backslash d})}{q(x^d|\mathbf{x}^{\backslash d})}$ whose rationale is Brook's Lemma [5, 49].

### 2.2 Graph Generation and Notations

We study the graphs with *categorical* node and edge attributes. A graph with $n$ nodes is represented by its edge type matrix and node type vector: $\mathcal{G} = (\mathbf{E}, \mathbf{F})$, where $\mathbf{E} = (e^{(i,j)})_{i,j \in \mathbb{N}_{\leq n}^+} \in \{1, \ldots, a + 1\}^{n \times n}$, $\mathbf{F} = (f^i)_{i \in \mathbb{N}_{\leq n}^+} \in \{1, \ldots, b\}^n$, $a$ and $b$ are the numbers of node and edge types, respectively. Notably, the absence of an edge is viewed as a special edge type, so there are $(a + 1)$ edge types in total. The problem we study is graph generation where $N$ graphs $\{\mathcal{G}^i\}_{i \in \mathbb{N}_{\leq N}^+}$ from an inaccessible graph data distribution $\mathfrak{G}$ are given and we aim to generate $M$ graphs $\{\mathcal{G}^i\}_{i \in \mathbb{N}_{\leq M}^+}$ from $\mathfrak{G}$.

## 3 Method

This section presents the proposed discrete-state continuous-time graph diffusion model, DISCO whose overview is Figure 2. Section 3.1 introduces the necessity to factorize the diffusion process and Section 3.2 details the forward process. Our training objective and its connection to sampling are introduced in Sections 3.3 and 3.4, respectively. Last but not least, a specific neural architecture of the graph-to-graph backbone model and its properties regarding the permutation of node ordering are introduced in Sections 3.5 and 3.6, respectively. *All proofs are in Appendix.*

### 3.1 Factorized Discrete Graph Diffusion Process

The number of possible states of an $n$-node graph is $(a + 1)^{n^2} \times b^n$ which is intractably large. Thus, we follow existing discrete models [2, 8, 71, 73] and formulate the forward processes on every node/edge to be independent. Mathematically, the forward diffusion process for $s < t$ is factorized as

$$q_{t|s}(\mathcal{G}_t|\mathcal{G}_s) = \prod_{i,j=1}^{n} q_{t|s}(e_t^{(i,j)}|e_s^{(i,j)}) \prod_{i=1}^{n} q_{t|s}(f_t^i|f_s^i) \qquad (2)$$

where the edge type transition probabilities $\{q_{t|s}(e_t^{(i,j)}|e_s^{(i,j)})\}_{i,j \in \mathbb{N}_{\leq n}^+}$ and node type transition probabilities $\{q_{t|s}(f_t^i|f_s^i)\}_{i \in \mathbb{N}_{\leq n}^+}$ are characterized by their forward rate matrices $\{\mathbf{R}_t^{(i,j)}\}_{i,j \in \mathbb{N}_{\leq n}^+}$ and $\{\mathbf{R}_t^i\}_{i \in \mathbb{N}_{\leq n}^+}$, respectively. The forward processes, i.e., the forward rate matrices in our context, are predefined, which will be introduced in Section 3.2. Given the factorization of forward transition probability in Eq. (2), a question is raised: *what is the corresponding factorization of the forward rate matrix ($\mathbf{R}_t$) and the reverse rate matrix ($\tilde{\mathbf{R}}_t$)?* Remark 3.1 shows such a factorization.

---

**Algorithm 1** Training of DISCO

---

1: A training graph $\mathcal{G}_0 = (\{f_0^i\}, \{e_0^{(i,j)}\})$ is given.

2: Sample $t \sim \mathcal{U}_{(0,T)}$; sample $\mathcal{G}_t$ based on transition probabilities $q_{t|0}(f_t = v|f_0 = u) = (e^{\int_0^t \beta(s)\mathbf{R}_f \, ds})_{uv}$ and $q_{t|0}(e_t = v|e_0 = u) = (e^{\int_0^t \beta(s)\mathbf{R}_e \, ds})_{uv}$, given $\mathcal{G}_0 = (\{f_0^i\}, \{e_0^{(i,j)}\})$.

3: Predict the clean graph $\hat{\mathcal{G}}_0 = (\{\hat{f}_0^i\}, \{\hat{e}_0^{(i,j)}\}) \leftarrow \left( \{p_{0|t}^\theta(f^i|\mathcal{G}_t)\}, \{p_{0|t}^\theta(e^{(i,j)}|\mathcal{G}_t)\} \right)$, given $\mathcal{G}_t$.

4: Compute cross-entropy loss between $\mathcal{G}_0$ and $\hat{\mathcal{G}}_0$ based on Eq. (6) and update $\theta$.

---

*Remark* 3.1. (Factorization of rate matrices, extended from Proposition 3 of [8]) Given the factorized forward process Eq. (2), the overall rate matrices are factorized as

$$\mathbf{R}_t(\bar{\mathcal{G}}, \mathcal{G}) = \sum_i A_t^i + \sum_{i,j} B_t^{(i,j)} \tag{3}$$

$$\tilde{\mathbf{R}}_t(\mathcal{G}, \bar{\mathcal{G}}) = \sum_i A_t^i \sum_{f_0^i} \frac{q_{t|0}(\bar{f}^i|f_0^i)}{q_{t|0}(f^i|f_0^i)} q_{0|t}(f_0^i|\mathcal{G}) + \sum_{i,j} B_t^{(i,j)} \sum_{e_0^{(i,j)}} \frac{q_{t|0}(\bar{e}^{(i,j)}|e_0^{(i,j)})}{q_{t|0}(e^{(i,j)}|e_0^{(i,j)})} q_{0|t}(e_0^{(i,j)}|\mathcal{G}) \tag{4}$$

where $A_t^i = \mathbf{R}_t^i(\bar{f}^i, f^i)\delta_{\bar{\mathcal{G}}\backslash \bar{f}^i, \mathcal{G}\backslash f^i}$, $B_t^{(i,j)} = \mathbf{R}_t^{(i,j)}(\bar{e}^{(i,j)}, e^{(i,j)})\delta_{\bar{\mathcal{G}}\backslash \bar{e}^{(i,j)}, \mathcal{G}\backslash e^{(i,j)}}$, the operator $\delta_{\bar{\mathcal{G}}\backslash \bar{f}^i, \mathcal{G}\backslash f^i}$ (or $\delta_{\bar{\mathcal{G}}\backslash \bar{e}^{(i,j)}, \mathcal{G}\backslash e^{(i,j)}}$) checks whether two graphs $\bar{\mathcal{G}}$ and $\mathcal{G}$ are exactly the same except for node $i$ (or the edge between nodes $i$ and $j$).

Note that this factorization itself is not our contribution but a necessary part of our framework, so we mention it here for completeness. Its full derivation is in Appendix - Section A. Next, we detail the design of forward rate matrices.

## 3.2 Forward Process

A proper choice of the forward rate matrices $\{\mathbf{R}_t^{(i,j)}\}_{i,j\in\mathbb{N}_{\leq n}^+}$ and $\{\mathbf{R}_t^i\}_{i\in\mathbb{N}_{\leq n}^+}$ is important because (1) the probability distributions of node and edge types, $\{q(f_t^i)\}_{i\in\mathbb{N}_{\leq n}^+}$ and $\{q(e_t^{(i,j)})\}_{i,j\in\mathbb{N}_{\leq n}^+}$, should converge to their reference distributions within $[0,T]$ and (2) the reference distributions should be easy to sample (e.g., uniform distribution). We follow [8] to formulate $\mathbf{R}_t^{(i,j)} = \beta(t)\mathbf{R}_e^{(i,j)}, \forall i,j$ and $\mathbf{R}_t^i = \beta(t)\mathbf{R}_f^i, \forall i$, where $\beta(t)$ is a corruption schedule, $\{\mathbf{R}_e^{(i,j)}\}$ and $\{\mathbf{R}_f^i\}$ are the base rate matrices. For brevity, we set all the nodes/edges to share a common node/edge rate matrix, i.e., $\mathbf{R}_e^{(i,j)} = \mathbf{R}_e$ and $\mathbf{R}_f^i = \mathbf{R}_f, \forall i,j$. Then, the forward transition probability for all the nodes and edges are $q_{t|0}(f_t = v|f_0 = u) = (e^{\int_0^t \beta(s)\mathbf{R}_f \, ds})_{uv}$ and $q_{t|0}(e_t = v|e_0 = u) = (e^{\int_0^t \beta(s)\mathbf{R}_e \, ds})_{uv}$, respectively. We omit the superscript $i$ (or $(i,j)$) because the transition probability is shared by all the nodes (or edges). The detailed derivation of the above analytic forward transition probability is provided in Appendix - Section B.

For categorical data, a reasonable reference distribution is a uniform distribution, i.e., $\pi_f = \frac{1}{b}$ for nodes and $\pi_e = \frac{1}{a+1}$ for edges. In addition, inspired by [73], we find that node and edge marginal distributions $\mathbf{m}_f$ and $\mathbf{m}_e$ are good choices as the reference distributions. Concretely, an empirical estimation of $\mathbf{m}_f$ and $\mathbf{m}_e$ is to count the number of node/edge types and normalize them. The following proposition shows how to design the rate matrices to guide the forward process to converge to uniform and marginal distributions.

**Proposition 3.2.** *The forward processes for nodes and edges converge to uniform distributions if $\mathbf{R}_f = \mathbf{1}\mathbf{1}^\top - b\mathbf{I}$ and $\mathbf{R}_e = \mathbf{1}\mathbf{1}^\top - (a+1)\mathbf{I}$; they converge to marginal distributions $\mathbf{m}_f$ and $\mathbf{m}_e$ if $\mathbf{R}_f = \mathbf{1}\mathbf{m}_f^\top - \mathbf{I}$ and $\mathbf{R}_e = \mathbf{1}\mathbf{m}_e^\top - \mathbf{I}$. $\mathbf{1}$ is an all-one vector and $\mathbf{I}$ is an identity matrix.*

Regarding the selection of $\beta(t)$, we follow [23, 70, 8] and set $\beta(t) = \alpha\gamma^t \log(\gamma)$ for a smooth change of the rate matrix. $\alpha$ and $\gamma$ are hyperparameters. Detailed settings are in Appendix F.3.

### 3.3 Parameterization and Optimization Objective

Next, we introduce the estimation of the reverse process from its motivation. The reverse process is essentially determined by the reverse rate matrix $\tilde{\mathbf{R}}_t$ in Eq. (4), whose computation needs $q_{0|t}(f_0^i|\mathcal{G})$ and $q_{0|t}(e_0^{(i,j)}|\mathcal{G})$, $\forall i,j$; their exact estimation is expensive because according to Bayes' rule, $p_t(\mathcal{G})$ is needed, whose computation needs to enumerate all the given graphs: $p_t(\mathcal{G}) = \sum_{\mathcal{G}_0} q_{t|0}(\mathcal{G}|\mathcal{G}_0)\pi_{\mathtt{data}}(\mathcal{G}_0)$.

Thus, we propose parameterizing the reverse transition probabilities via a neural network $\theta$ whose specific architecture is introduced in Section 3.5. The terms $\{q_{0|t}(f_0^i|\mathcal{G})\}_{i\in\mathbb{N}_{\leq n}^+}$ and $\{q_{0|t}(e_0^{(i,j)}|\mathcal{G})\}_{i,j\in\mathbb{N}_{\leq n}^+}$ in Eq. (4) are replaced with the parameterized $\{p_{0|t}^\theta(f^i|\mathcal{G})\}_{i\in\mathbb{N}_{\leq n}^+}$ and $\{p_{0|t}^\theta(e^{(i,j)}|\mathcal{G})\}_{i,j\in\mathbb{N}_{\leq n}^+}$. Thus, a parameterized reverse rate matrix $\tilde{\mathbf{R}}_{\theta,t}(\mathcal{G},\bar{\mathcal{G}})$ is represented as $\tilde{\mathbf{R}}_{\theta,t}(\mathcal{G},\bar{\mathcal{G}}) = \sum_i \tilde{\mathbf{R}}_{\theta,t}^i(f^i,\bar{f}^i) + \sum_{i,j} \tilde{\mathbf{R}}_{\theta,t}^{(i,j)}(e^{(i,j)},\bar{e}^{(i,j)})$ where $\tilde{\mathbf{R}}_{\theta,t}^i(f^i,\bar{f}^i) = A_t^i \sum_{f_0^i} \frac{q_{t|0}(\bar{f}^i|f_0^i)}{q_{t|0}(f^i|f_0^i)} p_{0|t}^\theta(f_0^i|\mathcal{G})$, $\tilde{\mathbf{R}}_{\theta,t}^{(i,j)}(e^{(i,j)},\bar{e}^{(i,j)}) = B_t^{(i,j)} \sum_{e_0^{(i,j)}} \frac{q_{t|0}(\bar{e}^{(i,j)}|e_0^{(i,j)})}{q_{t|0}(e^{(i,j)}|e_0^{(i,j)})} p_{0|t}^\theta(e_0^{(i,j)}|\mathcal{G})$, and the remaining notations are the same as Eq. (4). Note that all the terms $\{p_{0|t}^\theta(f^i|\mathcal{G})\}_{i\in\mathbb{N}_{\leq n}^+}$ and $\{p_{0|t}^\theta(e^{(i,j)}|\mathcal{G})\}_{i,j\in\mathbb{N}_{\leq n}^+}$ can be viewed together as a graph-to-graph mapping $\theta : \mathfrak{G} \mapsto \mathfrak{G}$, whose input is the noisy graph $\mathcal{G}_t$ and its output is the predicted clean graph probabilities, concretely, the node/edge type probabilities of all the nodes and edges.

Intuitively, the discrepancy between the groundtruth $\tilde{\mathbf{R}}_t$ (from Eq. (4)) and the parametric $\tilde{\mathbf{R}}_{\theta,t}$ should be small. Theorem 3.3 establishes a cross-entropy (CE)-based upper bound of such a discrepancy, where the estimated probability vectors (sum is 1) are notated as $\hat{f}_0^i = [p_{0|t}^\theta(f^i = 1|\mathcal{G}_t),\ldots,p_{0|t}^\theta(f^i = b|\mathcal{G}_t)]^\top \in [0,1]^b$ and $\hat{e}_0^{(i,j)} = [p_{0|t}^\theta(e^{(i,j)} = 1|\mathcal{G}_t),\ldots,p_{0|t}^\theta(e^{(i,j)} = a + 1|\mathcal{G}_t)]^\top \in [0,1]^{a+1}$.

**Theorem 3.3** (Approximation error). *for $\mathcal{G} \neq \bar{\mathcal{G}}$*

$$\left|\tilde{\mathbf{R}}_t(\mathcal{G},\bar{\mathcal{G}}) - \tilde{\mathbf{R}}_{\theta,t}(\mathcal{G},\bar{\mathcal{G}})\right|^2 \leq C_t + C_t^{\mathtt{node}}\mathbb{E}_{\mathcal{G}_0}q_{t|0}(\mathcal{G}|\mathcal{G}_0)\sum_i \mathcal{L}_{\mathrm{CE}}\big(\mathtt{One\text{-}Hot}(f_0^i), \hat{f}_0^i\big)$$

$$+ C_t^{\mathtt{edge}}\mathbb{E}_{\mathcal{G}_0}q_{t|0}(\mathcal{G}|\mathcal{G}_0)\sum_{i,j}\mathcal{L}_{\mathrm{CE}}\big(\mathtt{One\text{-}Hot}(e_0^{(i,j)}), \hat{e}_0^{(i,j)}\big) \tag{5}$$

*where $C_t$, $C_t^{\mathtt{node}}$, and $C_t^{\mathtt{edge}}$ are constants independent on $\theta$ but dependent on $t$, $\mathcal{G}$, and $\bar{\mathcal{G}}$; $\mathtt{One\text{-}Hot}$ transforms $f_0^i$ and $e_0^{(i,j)}$ into one-hot vectors.*

The bound in Theorem 3.3 is tight, i.e., the right-hand side of Eq. (5) is 0, whenever $\hat{f}_0^i = q_{0|t}(f_0^i|\mathcal{G}_t), \forall i$ and $\hat{e}_0^{(i,j)} = q_{0|t}(e_0^{(i,j)}|\mathcal{G}_t), \forall i,j$. Guided by Theorem 3.3, we (1) take expectation of $t$ by sampling $t$ from a uniform distribution $t \sim \mathcal{U}_{(0,T)}$ and (2) simplify the right-hand side of Eq. (5) by using the unweighted CE loss as our training objective:

$$\min_\theta T\mathbb{E}_t\mathbb{E}_{\mathcal{G}_0}\mathbb{E}_{q_{t|0}(\mathcal{G}_t|\mathcal{G}_0)}\left[\sum_i \mathcal{L}_{\mathrm{CE}}(\mathtt{One\text{-}Hot}(f_0^i), \hat{f}_0^i) + \sum_{i,j}\mathcal{L}_{\mathrm{CE}}(\mathtt{One\text{-}Hot}(e_0^{(i,j)}), \hat{e}_0^{(i,j)})\right] \tag{6}$$

A step-by-step training algorithm is in Algorithm 1. Note that the above CE loss has been used in some diffusion models (e.g., [2, 8]) but lacks a good motivation, especially in the continuous-time setting. We motivate it based on the rate matrix discrepancy, as a unique contribution of this paper.

### 3.4 Sampling Reverse Process

Given the parametric reverse rate matrix $\tilde{\mathbf{R}}_{\theta,t}(\mathcal{G},\bar{\mathcal{G}})$, the graph generation process can be implemented by two steps: (1) sampling the reference distribution $\pi_{\mathtt{ref}}$ (i.e., $\pi_f$ for nodes and $\pi_e$ for edges) and (2) numerically simulating the CTMC from time $T$ to 0. The exact simulation of a CTMC has been studied for a long time, e.g., [16, 17, 1]. However, their simulation strategies only allow one transition (e.g., one edge/node type change) per step, which is highly inefficient for graphs as the number of nodes and edges is typically large; once a(n) node/edge is updated, $\tilde{\mathbf{R}}_{\theta,t}$ requires recomputation. A

practical approximation is to assume $\tilde{\mathbf{R}}_{\theta,t}$ is fixed during a time interval $[t-\tau,t]$, i.e., delaying the happening of transitions in $[t-\tau,t]$ and triggering them all together at the time $t-\tau$; this strategy is also known as $\tau$-leaping [18, 8, 71], and DISCO adopts it.

We elaborate on $\tau$-leaping for transitions of node types; the transitions of edge types are similar. The rate matrix of the $i$-th node is fixed as $\tilde{\mathbf{R}}_{\theta,t}^i(f^i,\bar{f}^i) = \mathbf{R}_t^i(\bar{f}^i,f^i)\sum_{f_0^i}\frac{q_{t|0}(\bar{f}^i|f_0^i)}{q_{t|0}(f^i|f_0^i)}p_{0|t}^\theta(f^i|\mathcal{G}_t)$, during $[t-\tau,t]$. According to the definition of rate matrix, in $[t-\tau,t]$, the number of transitions from $f^i$ to $\bar{f}^i$, namely $J_{f^i,\bar{f}^i}$, follows the Poisson distribution, i.e., $J_{f^i,\bar{f}^i}\sim\text{Poisson}(\tau\tilde{\mathbf{R}}_{\theta,t}^i(f^i,\bar{f}^i))$. For categorical data (e.g., node type), multiple transitions in $[t-\tau,t]$ are invalid and meaningless. In other words, for the $i$-th node, if the total number of transitions $\sum_{\bar{f}^i}J_{f^i,\bar{f}^i} > 1$, $f^i$ keeps unchanged in $[t-\tau,t]$; otherwise, if $\sum_{\bar{f}^i}J_{f^i,\bar{f}^i} = 1$ and $J_{f^i,s} = 1$, i.e., there is exact 1 transition, $f^i$ jumps to $s$. A step-by-step sampling algorithm (Algorithm 2) is in Appendix.

*Remark* 3.4. The sampling error of $\tau$-leaping is linear to $C_{\text{err}}$ [8], the approximation error of the reverse rates: $\sum_{\mathcal{G}\neq\bar{\mathcal{G}}}\left|\tilde{\mathbf{R}}_t(\mathcal{G},\bar{\mathcal{G}}) - \tilde{\mathbf{R}}_{\theta,t}(\mathcal{G},\bar{\mathcal{G}})\right| \leq C_{\text{err}}$. Interested readers are referred to Theorem 1 from [8]. Our Theorem 3.3 shows the connection between our training loss and $C_{\text{err}}$, which further verifies the correctness of our training loss.

## 3.5 Model Instantiation

As mentioned in Section 3.3, the parametric backbone $p_{0|t}^\theta(\mathcal{G}_0|\mathcal{G}_t)$ is a graph-to-graph mapping whose input is the noisy graph $\mathcal{G}_t$ and its output is the predicted denoised graph $\mathcal{G}_0$. There exists a broad range of neural network architectures. Notably, DiGress [73] uses a graph Transformer (GT) as $p_{0|t}^\theta$, a decent reference for our continuous-time framework. We name our model with the GT backbone as DISCO-GT and its detailed configuration is in Appendix F.3. The main advantage of the GT is its long-range interaction thanks to the complete self-attention graph; however, the architecture is very complex and includes multi-head self-attention modules, leading to expensive computation.

Beyond GTs, in this paper, we posit that a regular message-passing neural network (MPNN) [19] should be a promising choice for $p_{0|t}^\theta(\mathcal{G}_0|\mathcal{G}_t)$. It is recognized that the MPNNs' expressiveness might not be as good as GTs' [33, 7], e.g., in terms of long-range interactions. However, in our setting, the absence of an edge is viewed as a special type of edge and the whole graph is complete; therefore, such a limitation of MPNN is naturally mitigated, which is verified by our empirical evaluations.

Concretely, an MPNN-based graph-to-graph mapping is presented as follows, and DISCO with MPNN backbone is named DISCO-MPNN. Given a graph $\mathcal{G} = (\mathbf{E},\mathbf{F})$, where $\mathbf{E} \in \{1,\ldots,a,a+1\}^{n\times n}$, $\mathbf{F} \in \{1,\ldots,b\}^n$, we first transform both the matrix $\mathbf{E}$ and $\mathbf{F}$ into one-hot embeddings $\mathbf{E}_{\text{OH}} \in \{0,1\}^{n\times n\times(a+1)}$ and $\mathbf{F}_{\text{OH}} \in \{0,1\}^{n\times b}$. Then, some auxiliary features (e.g., the # of specific motifs) are extracted: $\mathbf{F}_{\text{aux}},\mathbf{y}_{\text{aux}} = \text{Aux}(\mathbf{E}_{\text{OH}})$ to overcome the expressiveness limitation of MPNNs [11]. Here $\mathbf{F}_{\text{aux}}$ and $\mathbf{y}_{\text{aux}}$ are the node and global auxiliary features, respectively. Note that a similar auxiliary feature engineering is also applied in DiGress [73]. More details about the Aux can be found in Appendix E. Then, three multi-layer perceptrons (MLPs) are used to map node features $\mathbf{F}_{\text{OH}} \oplus \mathbf{F}_{\text{aux}}$, edge features $\mathbf{E}_{\text{OH}}$, and global features $\mathbf{y}_{\text{aux}}$ into a common hidden space as $\mathbf{F}_{\text{hidden}} = \text{MLP}(\mathbf{F}_{\text{OH}} \oplus \mathbf{F}_{\text{aux}})$, $\mathbf{E}_{\text{hidden}} = \text{MLP}(\mathbf{E}_{\text{OH}})$, $\mathbf{y}_{\text{hidden}} = \text{MLP}(\mathbf{y}_{\text{aux}})$, where $\oplus$ is a concatenation operator. The following formulas present the update of node embeddings (e.g., $\mathbf{r}^i = \mathbf{F}(i,:)$), edge embedding (e.g., $\mathbf{r}^{(i,j)} = \mathbf{E}(i,j,:)$), and global embedding $\mathbf{y}$ in an MPNN layer, where we omit the subscript hidden if it does not cause ambiguity:

$$\mathbf{r}^i \leftarrow \text{FiLM}\left(\text{FiLM}\left(\mathbf{r}^i,\text{MLP}\left(\sum_{j=1}^n \mathbf{r}^{(j,i)}/n\right)\right),\mathbf{y}\right),\quad \mathbf{r}^{(i,j)} \leftarrow \text{FiLM}(\text{FiLM}(\mathbf{r}^{(i,j)},\mathbf{r}^i\odot\mathbf{r}^j),\mathbf{y}),\quad (7)$$

$$\mathbf{y} \leftarrow \mathbf{y} + \text{PNA}(\{\mathbf{r}^i\}_{i=1}^n) + \text{PNA}(\{\mathbf{r}^{(i,j)}\}_{i,j=1}^n). \quad (8)$$

The edge embeddings are aggregated by mean pooling (i.e., $\sum_{j=1}^n \mathbf{r}^{(j,i)}/n$); the node pair embeddings are passed to edges by Hadamard product (i.e., $\mathbf{r}^i\odot\mathbf{r}^j$); edge/node embeddings are merged to the global embedding $\mathbf{y}$ via the PNA module [12]; Some FiLM modules [57] are used for the interaction between node/edge/global embeddings. More details about the PNA and FiLM are in Appendix E. In this paper, we name Eqs. (7) and (8) on all nodes/edges together as an MPNN layer, $\mathbf{F},\mathbf{E},\mathbf{y} \leftarrow \text{MPNN}(\mathbf{F},\mathbf{E},\mathbf{y})$. Stacking multiple MPNN layers leads to larger model capacity.

Finally, two readout MLPs are used to project the node/edge embeddings into input dimensions, $\text{MLP}(\mathbf{F}) \in \mathbb{R}^{n \times b}$ and $\text{MLP}(\mathbf{E}) \in \mathbb{R}^{n \times n \times (a+1)}$, which are output after wrapped with `softmax`.

Both the proposed MPNN and the GT from DiGress [73] use the PNA and FiLM to merge embeddings, but MPNN does not have multi-head self-attention layers so that the computation overhead is lower.

### 3.6 Permutation Equivariance and Invariance

Reordering the nodes keeps the property of a given graph, which is known as permutation invariance. In addition, for a given function if its input is permuted and its output is permuted accordingly, such a behavior is known as permutation equivariance. In this subsection, we analyze permutation-equivariance/invariance of the (1) diffusion framework (Lemmas 3.5, 3.6, and 3.7), (2) sampling density (Theorem 3.8), and (3) training loss (Theorem 3.9).

**Lemma 3.5** (Permutation-equivariant layer)**.** *The proposed MPNN layer (Eqs. (7) and (8)) is permutation-equivariant.*

The auxiliary features from the `Aux` are also permutation-equivariant (see Appendix E). Thus, the whole MPNN-based backbone $p_{0|t}^{\theta}$ is permutation-equivariant. Note that the GT-based backbone from DiGress [73] is also permutation-equivariant whose proof is omitted as it is not our contribution. Next, we show the permutation invariance of the rate matrices.

**Lemma 3.6** (Permutation-invariant rate matrices)**.** *The forward rate matrix of* DISCO *is permutation-invariant if it is factorized as Eq. (3). The parametric reverse rate matrix of* DISCO *($\tilde{\mathbf{R}}_{\theta,t}$) is permutation-invariant whenever the graph-to-graph backbone $p_{0|t}^{\theta}$ is permutation-equivariant.*

**Lemma 3.7** (Permutation-invariant transition probability)**.** *For CTMC satisfying the Kolmogorov forward equation (Eq. (1)), if the rate matrix is permutation-invariant (i.e., $\mathbf{R}_t(\mathbf{x}_i, \mathbf{x}_j) = \mathbf{R}_t(\mathcal{P}(\mathbf{x}_i), \mathcal{P}(\mathbf{x}_j))$), the transition probability is permutation-invariant (i.e., $q_{t|s}(\mathbf{x}_t | \mathbf{x}_s) = q_{t|s}(\mathcal{P}(\mathbf{x}_t) | \mathcal{P}(\mathbf{x}_s))$, where $\mathcal{P}$ is a permutation.*

Based on Lemmas 3.6 and 3.7, DISCO's parametric reverse transition probability is permutation-invariant. The next theorem shows the permutation-invariance of the sampling probability.

**Theorem 3.8** (Permutation-invariant sampling probability)**.** *If both the reference distribution $\pi_{\texttt{ref}}$ and the reverse transition probability are permutation-invariant, the parametric sampling distribution $p_0^{\theta}(\mathcal{G}_0)$ is permutation-invariant.*

In addition, the next theorem shows the permutation invariance of the training loss.

**Theorem 3.9** (Permutation-invariant training loss)**.** *The proposed training loss Eq. (6) is invariant to any permutation of the input graph $\mathcal{G}_0$ if $p_{0|t}^{\theta}$ is permutation-equivariant.*

## 4 Experiments

This section includes: an effectiveness evaluation on plain graphs (Section 4.1) and molecule graphs (Section 4.2), an efficiency study (Section 4.3), and an ablation study (Section 4.4). Detailed settings (Sections F.1-F.3), additional effectiveness evaluation (Sections F.4, additional ablation study (Section F.5), convergence study (Section F.6), and visualization (Section F.7) are in Appendix. Our code is released [3].

### 4.1 Plain Graph Generation

**Datasets and metrics.** Datasets SBM, Planar [51], and Community [82] are used. The relative squared Maximum Mean Discrepancy (MMD) for degree distributions (Deg.), clustering coefficient distributions (Clus.), and orbit counts (Orb.) distributions (the number of occurrences of substructures with 4 nodes), Uniqueness(%), Novelty(%), and Validity(%) are chosen as metrics. Details about the datasets, metrics, baselines (Section F.2.2), and results on Community (Table 8) are in Appendix.

**Results.** Table 1 shows the effectiveness evaluation on SBD and Planar from which we observe:

---
[3]`https://github.com/pricexu/DisCo`

Table 1: Performance (mean±std) on SBM and Planar datasets.

| Dataset | Model | Deg.↓ | Clus.↓ | Orb.↓ | Unique ↑ | Novel ↑ | Valid ↑ |
|---|---|---|---|---|---|---|---|
| SBM | GraphRNN [82] | 6.9 | 1.7 | 3.1 | **100.0** | **100.0** | 5.0 |
| | GRAN [42] | 14.1 | 1.7 | 2.1 | **100.0** | **100.0** | 25.0 |
| | GG-GAN [37] | 4.4 | 2.1 | 2.3 | **100.0** | **100.0** | 0.0 |
| | MolGAN [9] | 29.4 | 3.5 | 2.8 | 95.0 | **100.0** | 10.0 |
| | SPECTRE [51] | 1.9 | 1.6 | **1.6** | **100.0** | **100.0** | 52.5 |
| | ConGress [73] | 34.1 | 3.1 | 4.5 | 0.0 | 0.0 | 0.0 |
| | DiGress [73] | 1.6 | 1.5 | 1.7 | **100.0** | **100.0** | **67.5** |
| | DISCO-MPNN | $1.8_{\pm0.2}$ | $\mathbf{0.8}_{\pm\mathbf{0.1}}$ | $2.7_{\pm0.4}$ | $\mathbf{100.0}_{\pm\mathbf{0.0}}$ | $\mathbf{100.0}_{\pm\mathbf{0.0}}$ | $41.9_{\pm2.2}$ |
| | DISCO-GT | $\mathbf{0.8}_{\pm\mathbf{0.2}}$ | $\mathbf{0.8}_{\pm\mathbf{0.4}}$ | $2.0_{\pm0.5}$ | $\mathbf{100.0}_{\pm\mathbf{0.0}}$ | $\mathbf{100.0}_{\pm\mathbf{0.0}}$ | $66.2_{\pm1.4}$ |
| Planar | GraphRNN [82] | 24.5 | 9.0 | 2508.0 | **100.0** | **100.0** | 0.0 |
| | GRAN [42] | 3.5 | 1.4 | 1.8 | 85.0 | 2.5 | **97.5** |
| | GG-GAN [37] | 315.0 | 8.3 | 2062.6 | **100.0** | **100.0** | 0.0 |
| | MolGAN [9] | 4.5 | 10.2 | 2346.0 | 25.0 | **100.0** | 0.0 |
| | SPECTRE [51] | 2.5 | 2.5 | 2.4 | **100.0** | **100.0** | 25.0 |
| | ConGress [73] | 23.8 | 8.8 | 2590.0 | 0.0 | 0.0 | 0.0 |
| | DiGress [73] | 1.4 | **1.2** | **1.7** | **100.0** | **100.0** | 85.0 |
| | DISCO-MPNN | $1.4_{\pm0.3}$ | $1.4_{\pm0.4}$ | $6.4_{\pm1.6}$ | $\mathbf{100.0}_{\pm\mathbf{0.0}}$ | $\mathbf{100.0}_{\pm\mathbf{0.0}}$ | $33.8_{\pm2.7}$ |
| | DISCO-GT | $\mathbf{1.2}_{\pm\mathbf{0.5}}$ | $1.3_{\pm0.5}$ | $\mathbf{1.7}_{\pm\mathbf{0.7}}$ | $\mathbf{100.0}_{\pm\mathbf{0.0}}$ | $\mathbf{100.0}_{\pm\mathbf{0.0}}$ | $83.6_{\pm2.1}$ |

Table 2: Performance (mean±std%) on QM9 dataset. V., U., and N. mean Valid, Unique, and Novel.

| Model | Valid ↑ | V.U. ↑ | V.U.N. ↑ |
|---|---|---|---|
| CharacterVAE [20] | 10.3 | 7.0 | 6.3 |
| GrammarVAE[38] | 60.2 | 5.6 | 4.5 |
| GraphVAE [66] | 55.7 | 42.0 | 26.1 |
| GT-VAE [55] | 74.6 | 16.8 | 15.8 |
| Set2GraphVAE [72] | 59.9 | 56.2 | - |
| GG-GAN [37] | 51.2 | 24.4 | 24.4 |
| MolGAN [9] | 98.1 | 10.2 | 9.6 |
| SPECTRE [51] | 87.3 | 31.2 | 29.1 |
| GraphNVP [50] | 83.1 | 82.4 | - |
| GDSS [32] | 95.7 | 94.3 | - |
| EDGE [10] | 99.1 | **99.1** | - |
| ConGress [73] | 98.9 | 95.7 | 38.3 |
| DiGress [73] | 99.0 | 95.2 | 31.8 |
| GRAPHARM [36] | 90.3 | 86.3 | - |
| DISCO-MPNN | $98.9_{\pm0.7}$ | $98.7_{\pm0.5}$ | $\mathbf{68.7}_{\pm\mathbf{0.2}}$ |
| DISCO-GT | $\mathbf{99.3}_{\pm\mathbf{0.6}}$ | $98.9_{\pm0.6}$ | $56.2_{\pm0.4}$ |

- DISCO-GT can obtain competitive performance against the SOTA, DiGress, which is reasonable because both models share the graph Transformer backbone. Note that DiGress's performance in terms of Validity is not the statistics reported in the paper but from their latest model checkpoint [4]. In fact, we found it very hard for DiGress and DISCO-GT to learn to generate valid SBM/Planar graphs. These two datasets have only 200 graphs, but sometimes only after $> 10,000$ epochs training, the Validity percentage can be $> 50\%$. Additionally, DISCO-GT provides extra flexibility during sampling by adjusting the $\tau$. This is important: our models can still trade-off between the sampling efficiency and quality even after the model is trained and frozen.
- In general, DISCO-MPNN has competitive performance against DISCO-GT in terms of Deg., Clus., and Orb. However, its performance is worse compared to DISCO-GT in terms of Validity, which might be related to the different model expressiveness. Studying the graph-to-graph model expressiveness would be an interesting future direction, e.g., generating valid Planar graphs.

---

[4] https://github.com/cvignac/DiGress/blob/main/README.md

Table 3: Performance on MOSES. VAE, JT-VAE, and GraphINVENT have hard-coded rules to ensure high validity.

| Model | Valid ↑ | Unique ↑ | Novel ↑ | Filters ↑ | FCD ↓ | SNN ↑ | Scaf ↑ |
|---|---|---|---|---|---|---|---|
| VAE [21] | 97.7 | 98.8 | 69.5 | 99.7 | 0.57 | 0.58 | 5.9 |
| JT-VAE [29] | 100.0 | 100.0 | 99.9 | 97.8 | 1.00 | 0.53 | 10.0 |
| GraphINVENT [53] | 96.4 | 99.8 | N/A | 95.0 | 1.22 | 0.54 | 12.7 |
| ConGress [73] | 83.4 | 99.9 | 96.4 | 94.8 | 1.48 | 0.50 | 16.4 |
| DiGress [73] | 85.7 | 100.0 | 95.0 | 97.1 | 1.19 | 0.52 | 14.8 |
| DISCO-MPNN | 83.9 | 100.0 | 98.8 | 87.3 | 1.63 | 0.48 | 13.5 |
| DISCO-GT | 88.3 | 100.0 | 97.7 | 95.6 | 1.44 | 0.50 | 15.1 |

Table 4: Performance on GuacaMol. LSTM, NAGVAE, and MCTS are tailored for molecule datasets; ConGress, DiGress, and DISCO are general graph generation models.

| Model | Valid ↑ | Unique ↑ | Novel ↑ | KL div ↑ | FCD ↑ |
|---|---|---|---|---|---|
| LSTM [64] | 95.9 | 100.0 | 91.2 | 99.1 | 91.3 |
| NAGVAE [40] | 92.9 | 95.5 | 100.0 | 38.4 | 0.9 |
| MCTS [28] | 100.0 | 100.0 | 95.4 | 82.2 | 1.5 |
| ConGress [73] | 0.1 | 100.0 | 100.0 | 36.1 | 0.0 |
| DiGress [73] | 85.2 | 100.0 | 99.9 | 92.9 | 68.0 |
| DISCO-MPNN | 68.7 | 100.0 | 96.4 | 77.0 | 36.4 |
| DISCO-GT | 86.6 | 100.0 | 99.9 | 92.6 | 59.7 |

## 4.2 Molecule Graph Generation

**Dataset and metrics.** The datasets QM9 [62], MOSES [58], and GuacaMol [6] are chosen. For MOSES, metrics including Uniquess, Novelty, Validity, Filter, FCD, SNN, and Scaf are reported in Table 3. For QM9, metrics include Uniqueness, Novelty, and Validity. For GuacaMol, metrics include Valid, Unique, Novel, KL div, and FCD. Details about the datasets, metrics, and baseline methods are in Appendix F.2.3.

**Results.** Table 2 shows the performance on QM9 dataset. Our observation is consistent with the performance comparison on plain datasets: (1) DISCO-GT obtains slightly better or at least competitive performance against DiGress due to the shared graph-to-graph backbone, but our framework offers extra flexibility in the sampling process; (2) DISCO-MPNN obtains decent performance in terms of Validity, Uniqueness, and Novelty comparing with DISCO-GT.

Tables 3 and 4 show the performance on MOSES and GuacaMol which further verifies that (1) performance of DISCO-GT is on par with the SOTA general graph generative models, DiGress and (2) DISCO-MPNN has decent performance, but worse than DISCO-GT and DiGress.

## 4.3 Efficiency Study

A major computation bottleneck is the graph-to-graph backbone $p_{0|t}^\theta$, which is GT or MPNN. We compare the number of parameters, the forward and back-propagation time of GT and MPNN in Table 5. For a fair comparison, we set all the hidden dimensions of GT and MPNN as $256$ and the number of layers as $5$. We use the Community [82] dataset and set the batch size as $64$. Table 5 shows that GT has a larger capacity and more parameters at the expense of more expensive training.

Table 5: Efficiency comparison in terms of number of parameters, forward and backpropagation time (second/iteration).

| | GT | MPNN |
|---|---|---|
| # Parameters | $14 \times 10^6$ | $7 \times 10^6$ |
| Forward | 0.065 | 0.022 |
| Backprop. | 0.034 | 0.018 |

#### 4.4 Ablation Study

An ablation study on DISCO-GT for reference distributions (marginal vs. uniform), and sampling steps (1 to 500) is presented in Table 6. The number of sampling steps is $\mathrm{round}(\frac{1}{\tau})$ if $T = 1$. QM9 dataset is chosen. A similar ablation study on DISCO-MPNN is in Table 9 in Appendix. We observe that first, generally, the fewer sampling steps, the lower the generation quality. In some cases (e.g., the marginal distribution) with the sampling steps decreasing significantly (e.g., from 500 to 30), the performance degradation is still very slight, implying our method's high robustness in sampling steps. Second, the marginal reference distribution is better than the uniform distribution, consistent with the observation from DiGress [73].

Table 6: Ablation study (mean±std%) with GT backbone. V., U., and N. mean Valid, Unique, and Novel.

| Ref. Dist. | Steps | Valid ↑ | V.U. ↑ | V.U.N. ↑ |
|---|---|---|---|---|
| Marginal | 500 | $99.3_{\pm 0.6}$ | $98.9_{\pm 0.6}$ | $56.2_{\pm 0.4}$ |
| | 100 | $98.7_{\pm 0.5}$ | $98.5_{\pm 0.4}$ | $58.8_{\pm 0.4}$ |
| | 30 | $97.9_{\pm 1.2}$ | $97.6_{\pm 1.1}$ | $59.2_{\pm 0.8}$ |
| | 10 | $95.3_{\pm 1.9}$ | $94.8_{\pm 1.6}$ | $62.1_{\pm 0.9}$ |
| | 5 | $93.0_{\pm 1.7}$ | $92.4_{\pm 1.3}$ | $64.9_{\pm 1.1}$ |
| | 1 | $76.1_{\pm 2.3}$ | $73.9_{\pm 1.6}$ | $62.9_{\pm 1.8}$ |
| Uniform | 500 | $94.1_{\pm 0.9}$ | $92.9_{\pm 0.5}$ | $56.6_{\pm 0.4}$ |
| | 100 | $91.5_{\pm 1.0}$ | $90.3_{\pm 0.9}$ | $54.4_{\pm 1.2}$ |
| | 30 | $88.7_{\pm 1.6}$ | $86.9_{\pm 1.0}$ | $58.6_{\pm 2.1}$ |
| | 10 | $84.5_{\pm 2.3}$ | $80.4_{\pm 1.7}$ | $59.8_{\pm 1.8}$ |
| | 5 | $77.0_{\pm 2.5}$ | $69.9_{\pm 1.5}$ | $56.1_{\pm 3.5}$ |
| | 1 | $44.9_{\pm 3.1}$ | $35.1_{\pm 3.4}$ | $29.6_{\pm 2.5}$ |

## 5 Related Work

Diffusion models [80] can be interpreted from both the score-matching [69, 70] or the variational autoencoder perspective [23, 35, 34]. Pioneering efforts on diffusion generative modeling study the process in continuous-state [67, 23, 68] whose typical reference distribution is Gaussian. Beyond that, some efforts propose discrete-state models [24] to . E.g., D3PM [2] designs the discrete diffusion process by multiplication of transition matrices; $\tau$-LDR [8] generalizes D3PM by formulating a continuous-time Markov chain; [71] proposes a singleton conditional distribution-based objective for the continuous-time Markov chain-based model whose rationale is Brook's Lemma [5, 49].

Diffusion models are widely used in graph generation tasks [44, 13, 85, 86, 84, 15, 83, 14, 61, 74, 47, 79, 78, 59, 3] such as molecule design [65, 25, 27, 43]. Pioneering works such as EDP-GNN [56] and GDSS [32] diffuse graph data in a continuous state space [26]. DiscDDPM [22] is an early effort to modify the DDPM architecture into a discrete state. In addition, DiGress [73] is also a one-shot discrete-state diffusion model, followed by a very recent work MCD [46], both in the discrete-time setting. Beyond the above-mentioned efforts, DruM [31] proposes to mix the diffusion process. EDGE [10] proposes an interesting process: diffusing graphs into empty graphs. Besides, GRAPHARM [36] proposes an autoregressive graph diffusion model, and [45] applies the diffusion models for molecule property prediction tasks. In addition to the above-mentioned general graph diffusion models, there are many other task-tailored graph diffusion generative models [48, 30, 41, 60, 77, 76, 4, 75], which incorporate more in-depth domain expertise into the model design. Interested readers are referred to this survey [44].

## 6 Conclusion

This paper introduces the first discrete-state continuous-time graph diffusion generative model, DISCO. Our model effectively marries continuous-time Markov Chain formulation with the discrete nature of graph data, addressing the fundamental sampling limitation of prior models. DISCO's training objective is concise with a solid theoretical foundation. We also propose a simplified message-passing architecture to serve as the graph-to-graph backbone, which theoretically has desirable properties against permutation of node ordering and empirically demonstrates decent performance against existing graph generative models in tests on various datasets.

## Acknowledgments and Disclosure of Funding

ZX, RQ, ZZ, and HT are partially supported by NSF (2324770). The content of the information in this document does not necessarily reflect the position or the policy of the Government, and no official endorsement should be inferred. The U.S. Government is authorized to reproduce and distribute reprints for Government purposes notwithstanding any copyright notation here on.

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

# Appendix

The organization of this appendix is as follows

## A  Details of the Factorization of Rate Matrices

In this section, we detail the derivation of Remark 3.1, which is extended from the following Proposition 3 of [8].

**Proposition A.1** (Factorization of the rate matrix, Proposition 3 from [8])**.** *If the forward process factorizes as* $q_{t|s}(\mathbf{x}_t|\mathbf{x}_s) = \prod_{d=1}^{D} q_{t|s}(x_t^d|x_s^d), t > s$, *then the forward and reverse rates are of the form*

$$\mathbf{R}_t(\bar{\mathbf{x}}, \mathbf{x}) = \sum_{d=1}^{D} \mathbf{R}_t^d(\bar{x}^d, x^d) \delta_{\bar{\mathbf{x}} \backslash \bar{x}^d, \mathbf{x} \backslash x^d} \tag{9}$$

$$\tilde{\mathbf{R}}_t(\mathbf{x}, \bar{\mathbf{x}}) = \sum_{d=1}^{D} \mathbf{R}_t^d(\bar{x}^d, x^d) \delta_{\bar{\mathbf{x}} \backslash \bar{x}^d, \mathbf{x} \backslash x^d} \sum_{x_0^d} q_{0|t}(x_0^d|\mathbf{x}) \frac{q_{t|0}(\bar{x}^d|x_0^d)}{q_{t|0}(x^d|x_0^d)} \tag{10}$$

*where* $\delta_{\bar{\mathbf{x}} \backslash \bar{x}^d, \mathbf{x} \backslash x^d} = 1$ *when all dimensions except for* $d$ *are equal.*

As all the nodes and edges are categorical, applying the above proposition of all the nodes and edges leads to our Remark 3.1.

## B  Details of Forward Transition Probability

In this section, we present the derivation of the forward transition probability for nodes; the forward process for edges can be derived similarly. Note that this derivation has been mentioned in [8] for generic discrete cases; we graft it to the graph settings and include it here for completeness. The core derivation of the forward transition probability is to prove the following proposition.

**Proposition B.1** (Analytical forward process for commutable rate matrices, Proposition 10 from [8]). *if* $\mathbf{R}_t$ *and* $\mathbf{R}_{t'}$ *commute* $\forall t, t'$, $q_{t|0}(x_t = j|x_0 = i) = (e^{\int_0^t \mathbf{R}_s ds})_{ij}$

*Proof.* If $q_{t|0} = \exp\left(\int_0^t \mathbf{R}_s ds\right)$ is the forward transition probability matrix, it should satisfy the Kolmogorov forward equation $\frac{d}{dt} q_{t|0} = q_{t|0} \mathbf{R}_s$. The transition probability matrix

$$q_{t|0} = \sum_{k=0}^{\infty} \frac{1}{k!} \left(\int_0^t \mathbf{R}_s ds\right)^k, \tag{11}$$

and, based on the fact that $\mathbf{R}_t$ and $\mathbf{R}'_t$ commute $\forall t, t'$, its derivative is

$$\frac{d}{dt} q_{t|0} = \sum_{k=1}^{\infty} \frac{1}{(k-1)!} \left(\int_0^t \mathbf{R}_s ds\right)^{(k-1)} = q_{t|0} \mathbf{R}_t. \tag{12}$$

Thus, $q_{t|0} = \exp\left(\int_0^t \mathbf{R}_s ds\right)$ is the solution of Kolmogorov forward equation. $\square$

For the node $i$, if its forward rate matrix is set as $\mathbf{R}_t^i = \beta(t)\mathbf{R}_f$, we have $\mathbf{R}_t^i$ and $\mathbf{R}_{t'}^i$ commute, $\forall t, t'$. Thus, the transition probability for node $i$ is $q_{t|0}(f_t^i = v|f_0^i = u) = (e^{\int_0^t \beta(s)\mathbf{R}_f ds})_{uv}$. Based on similar derivation, we have the transition probability for the edge $(i, j)$ as $q_{t|0}(e_t^{(i,j)} = v|e_0^{(i,j)} = u) = (e^{\int_0^t \beta(s)\mathbf{R}_e ds})_{uv}$.

## C   Proofs

### C.1   Proof of Proposition 3.2

Proposition 3.2 claims the forward process converges to uniform distributions if $\mathbf{R}_f = \mathbf{1}\mathbf{1}^\top - b\mathbf{I}$ and $\mathbf{R}_e = \mathbf{1}\mathbf{1}^\top - (a+1)\mathbf{I}$ and it converges to marginal distributions $\mathbf{m}_f$ and $\mathbf{m}_e$ if $\mathbf{R}_f = \mathbf{1}\mathbf{m}_f^\top - \mathbf{I}$ and $\mathbf{R}_e = \mathbf{1}\mathbf{m}_e^\top - \mathbf{I}$.

*Proof.* If we formulate the rate matrices for nodes and edges as $\mathbf{R}_t^{(i,j)} = \beta(t)\mathbf{R}_e$, $\forall i, j$ and $\mathbf{R}_t^i = \beta(t)\mathbf{R}_f$, $\forall i$, every rate matrix is commutable for any time steps $t$ and $t'$. In the following content, we show the proof for the node rate matrix $\mathbf{R}_t^i = \beta(t)\mathbf{R}_f$; the converged distribution of edge can be proved similarly. Based on Proposition B.1, the transition probability matrix between time steps $t$ and $t + \Delta t$ is

$$q_{t+\Delta t|t} = \mathbf{I} + \int_t^{t+\Delta t} \beta(s)\mathbf{R}_f ds + O((\Delta t)^2) \tag{13}$$

$$\overset{(*)}{=} \mathbf{I} + \Delta t \beta(\xi)\mathbf{R}_f + O((\Delta t)^2), \tag{14}$$

where (*) is based on the Mean Value Theorem. If the high-order term $O((\Delta t)^2)$ is omitted and we short $\beta_{\Delta t} = \Delta t \beta(\xi)$, for $\mathbf{R}_f = \mathbf{1}\mathbf{1}^\top - b\mathbf{I}$, we have

$$q_{t+\Delta t|t} \approx \beta_{\Delta t}\mathbf{1}\mathbf{1}^\top + (1 - \beta_{\Delta t}b)\mathbf{I}, \tag{15}$$

which is the transition matrix of the uniform diffusion in the discrete-time diffusion models [67, 2]. Thus, with $T \to \infty$ and $q_{t+\Delta t|t}$ to the power of infinite, the converged distribution is a uniform distribution. Similarly, for $\mathbf{R}_f = \mathbf{1}\mathbf{m}_f^\top - \mathbf{I}$ the transition matrix is

$$q_{t+\Delta t|t} \approx \beta_{\Delta t}\mathbf{1}\mathbf{m}_f^\top + (1 - \beta_{\Delta t})\mathbf{I} \tag{16}$$

which is a generalized transition matrix of the 'absorbing state' diffusion [2]. The difference lies at for the 'absorbing state' diffusion [2], $\mathbf{m}_f$ is set as a one-hot vector for the absorbing state, and here we set it as the marginal distribution. Thus, with $T \to \infty$ and $q_{t+\Delta t|t}$ to the power of infinite, the converged distribution is a marginal distribution $\mathbf{m}_f$. $\square$

## C.2 Proof of Theorem 3.3

Theorem 3.3 says for $\mathcal{G} \neq \bar{\mathcal{G}}$,

$$\left| \tilde{\mathbf{R}}_t(\mathcal{G}, \bar{\mathcal{G}}) - \tilde{\mathbf{R}}_{\theta,t}(\mathcal{G}, \bar{\mathcal{G}}) \right|^2 \leq C_t + C_t^{\texttt{node}} \mathbb{E}_{\mathcal{G}_0} q_{t|0}(\mathcal{G}|\mathcal{G}_0) \sum_i \mathcal{L}_{\texttt{CE}}(\texttt{One-Hot}(f_0^i), \hat{f}_0^i)$$

$$+ C_t^{\texttt{edge}} \mathbb{E}_{\mathcal{G}_0} q_{t|0}(\mathcal{G}|\mathcal{G}_0) \sum_{i,j} \mathcal{L}_{\texttt{CE}}(\texttt{One-Hot}(e_0^{(i,j)}), \hat{e}_0^{(i,j)}) \quad (17)$$

where the node and edge estimated probability vector (sum is 1) is notated as $\hat{f}_0^i = [p_{0|t}^\theta(f^i = 1|\mathcal{G}_t), \ldots, p_{0|t}^\theta(f^i = b|\mathcal{G}_t)]^\top \in [0,1]^b$ and $\hat{e}_0^{(i,j)} = [p_{0|t}^\theta(e^{(i,j)} = 1|\mathcal{G}_t), \ldots, p_{0|t}^\theta(e^{(i,j)} = a + 1|\mathcal{G}_t)]^\top \in [0,1]^{a+1}$.

*Proof.*

$$\left| \tilde{\mathbf{R}}_t(\mathcal{G}, \bar{\mathcal{G}}) - \tilde{\mathbf{R}}_{\theta,t}(\mathcal{G}, \bar{\mathcal{G}}) \right| \quad (18)$$

$$= \left| \sum_i A_t^i \sum_{f_0^i} \frac{q_{t|0}(\bar{f}^i|f_0^i)}{q_{t|0}(f^i|f_0^i)} (q_{0|t}(f_0^i|\mathcal{G}) - p_{0|t}^\theta(f_0^i|\mathcal{G})) \right.$$

$$\left. + \sum_{i,j} B_t^{(i,j)} \sum_{e_0^{(i,j)}} \frac{q_{t|0}(\bar{e}^{(i,j)}|e_0^{(i,j)})}{q_{t|0}(e^{(i,j)}|e_0^{(i,j)})} (q_{0|t}(e_0^{(i,j)}|\mathcal{G}) - p_{0|t}^\theta(e_0^{(i,j)}|\mathcal{G})) \right| \quad (19)$$

$$\leq \left| \sum_i A_t^i \sum_{f_0^i} \frac{q_{t|0}(\bar{f}^i|f_0^i)}{q_{t|0}(f^i|f_0^i)} (q_{0|t}(f_0^i|\mathcal{G}) - p_{0|t}^\theta(f_0^i|\mathcal{G})) \right|$$

$$+ \left| \sum_{i,j} B_t^{(i,j)} \sum_{e_0^{(i,j)}} \frac{q_{t|0}(\bar{e}^{(i,j)}|e_0^{(i,j)})}{q_{t|0}(e^{(i,j)}|e_0^{(i,j)})} (q_{0|t}(e_0^{(i,j)}|\mathcal{G}) - p_{0|t}^\theta(e_0^{(i,j)}|\mathcal{G})) \right| \quad (20)$$

We check the first term of Eq. (20):

$$\left| \sum_i A_t^i \sum_{f_0^i} \frac{q_{t|0}(\bar{f}^i|f_0^i)}{q_{t|0}(f^i|f_0^i)} (q_{0|t}(f_0^i|\mathcal{G}) - p_{0|t}^\theta(f_0^i|\mathcal{G})) \right| \quad (21)$$

$$\leq \sum_i A_t^i \sup_{f_0^i} \left\{ \frac{q_{t|0}(\bar{f}^i|f_0^i)}{q_{t|0}(f^i|f_0^i)} \right\} \sum_{f_0^i} \left| q_{0|t}(f_0^i|\mathcal{G}) - p_{0|t}^\theta(f_0^i|\mathcal{G}) \right| \quad (22)$$

$$= \sum_i C_i \sum_{f_0^i} \left| q_{0|t}(f_0^i|\mathcal{G}) - p_{0|t}^\theta(f_0^i|\mathcal{G}) \right| \quad (23)$$

$$\stackrel{(*)}{\leq} \sum_i C_i \sqrt{2 \sum_{f_0^i} \left( C_{f_0^i} - q_{0|t}(f_0^i|\mathcal{G}) \log p_{0|t}^\theta(f_0^i|\mathcal{G}) \right)} \quad (24)$$

$$\stackrel{(**)}{\leq} C_1 \sqrt{\sum_i \sum_{f_0^i} \left( C_{f_0^i} - q_{0|t}(f_0^i|\mathcal{G}) \log p_{0|t}^\theta(f_0^i|\mathcal{G}) \right)} \quad (25)$$

$$= C_1 \sqrt{C_2 - \sum_i \sum_{f_0^i} q_{0|t}(f_0^i|\mathcal{G}) \log p_{0|t}^\theta(f_0^i|\mathcal{G})} \quad (26)$$

where $C_i = A_t^i \sup_{f_0^i} \left\{ \frac{q_{t|0}(\bar{f}^i|f_0^i)}{q_{t|0}(f^i|f_0^i)} \right\}$, $C_{f_0^i} = q_{0|t}(f_0^i|\mathcal{G}) \log q_{0|t}(f_0^i|\mathcal{G})$, (*) is based on the Pinsker's inequality, (**) is based on Cauchy–Schwarz inequality: $\sum_{i=1}^n \sqrt{x_i} \leq \sqrt{n \sum_{i=1}^n x_i}$, $C_1 = \sqrt{2n} \sup_i \{C_i\}$, $C_2 = \sum_i \sum_{f_0^i} C_{f_0^i}$. Next, the term $-\sum_i \sum_{f_0^i} q_{0|t}(f_0^i|\mathcal{G}) \log p_{0|t}^\theta(f_0^i|\mathcal{G})$ is equiva-

lent to:

$$-\sum_i \sum_{f_0^i} q_{0|t}(f_0^i|\mathcal{G}) \log p_{0|t}^\theta(f_0^i|\mathcal{G}) \tag{27}$$

$$= -\frac{1}{p_t(\mathcal{G})} \sum_i \sum_{f_0^i} p_{0,t}(f_0^i, \mathcal{G}) \log p_{0|t}^\theta(f_0^i|\mathcal{G}) \tag{28}$$

$$= -\frac{1}{p_t(\mathcal{G})} \sum_i \sum_{f_0^i} \sum_{\mathcal{G}_0(f_0^i)} p_{0,t}(\mathcal{G}_0, \mathcal{G}) \log p_{0|t}^\theta(f_0^i|\mathcal{G}) \tag{29}$$

$$= -\frac{1}{p_t(\mathcal{G})} \sum_i \sum_{f_0^i} \sum_{\mathcal{G}_0(f_0^i)} \pi_{\texttt{data}}(\mathcal{G}_0) q_{t|0}(\mathcal{G}|\mathcal{G}_0) \log p_{0|t}^\theta(f_0^i|\mathcal{G}) \tag{30}$$

$$= \frac{1}{p_t(\mathcal{G})} \sum_i \sum_{f_0^i} \sum_{\mathcal{G}_0(f_0^i)} \pi_{\texttt{data}}(\mathcal{G}_0) q_{t|0}(\mathcal{G}|\mathcal{G}_0) \mathcal{L}_{\texttt{CE}}(\texttt{One-Hot}(f_0^i), \hat{f}_0^i) \tag{31}$$

$$= \frac{1}{p_t(\mathcal{G})} \sum_{\mathcal{G}_0} \pi_{\texttt{data}}(\mathcal{G}_0) q_{t|0}(\mathcal{G}|\mathcal{G}_0) \sum_i \mathcal{L}_{\texttt{CE}}(\texttt{One-Hot}(f_0^i), \hat{f}_0^i) \tag{32}$$

$$= \frac{1}{p_t(\mathcal{G})} \mathbb{E}_{\mathcal{G}_0} q_{t|0}(\mathcal{G}|\mathcal{G}_0) \sum_i \mathcal{L}_{\texttt{CE}}(\texttt{One-Hot}(f_0^i), \hat{f}_0^i) \tag{33}$$

where $\sum_{\mathcal{G}_0(f_0^i)}$ marginalizing all the graphs at time 0 whose $i$-th node is $f_0^i$; $p_{0,t}(f_0^i, \mathcal{G})$ is the joint probability of a graph whose $i$-th node is $f_0^i$ at time 0 and it is $\mathcal{G}$ at time $t$; $p_{0,t}(\mathcal{G}_0, \mathcal{G})$ is the joint probability of a graph which is $\mathcal{G}_0$ at time 0 and it is $\mathcal{G}$ at time $t$. Plugging Eq. (33) into Eq. (26):

$$\left| \sum_i A_t^i \sum_{f_0^i} \frac{q_{t|0}(\bar{f}^i|f_0^i)}{q_{t|0}(f^i|f_0^i)} (q_{0|t}(f_0^i|\mathcal{G}) - p_{0|t}^\theta(f_0^i|\mathcal{G})) \right|$$

$$\leq C_1 \sqrt{C_2 + C_5 \mathbb{E}_{\mathcal{G}_0} q_{t|0}(\mathcal{G}|\mathcal{G}_0) \sum_i \mathcal{L}_{\texttt{CE}}(\texttt{One-Hot}(f_0^i), \hat{f}_0^i)} \tag{34}$$

where $C_5 = \frac{1}{p_t(\mathcal{G})}$. A similar analysis can be conducted about the second term of Eq. (20) and we directly present it here:

$$\left| \sum_{i,j} B_t^{(i,j)} \sum_{e_0^{(i,j)}} \frac{q_{t|0}(\bar{e}^{(i,j)}|e_0^{(i,j)})}{q_{t|0}(e^{(i,j)}|e_0^{(i,j)})} (q_{0|t}(e_0^{(i,j)}|\mathcal{G}) - p_{0|t}^\theta(e_0^{(i,j)}|\mathcal{G})) \right|$$

$$\leq C_3 \sqrt{C_4 + C_5 \mathbb{E}_{\mathcal{G}_0} q_{t|0}(\mathcal{G}|\mathcal{G}_0) \sum_{i,j} \mathcal{L}_{\texttt{CE}}(\texttt{One-Hot}(e_0^{(i,j)}), \hat{e}_0^{(i,j)})} \tag{35}$$

where $C_3 = \sqrt{2}n \sup_{i,j}\{C_{i,j}\}$, $C_4 = \sum_{i,j} \sum_{e_0^{(i,j)}} C_{e_0^{(i,j)}}$, $C_{i,j} = B_t^{(i,j)} \sup_{e_0^{(i,j)}} \left\{ \frac{q_{t|0}(\bar{e}^{(i,j)}|e_0^{(i,j)})}{q_{t|0}(e^{(i,j)}|e_0^{(i,j)})} \right\}$, $C_{e_0^{(i,j)}} = q_{0|t}(e_0^{(i,j)}|\mathcal{G}) \log q_{0|t}(e_0^{(i,j)}|\mathcal{G})$.

Plugging Eqs. (34) and (35) into Eq. (20), being aware that $C_1, C_2, C_3, C_4, C_5$ are all $t$-related:

$$\left| \tilde{\mathbf{R}}_t(\mathcal{G}, \bar{\mathcal{G}}) - \tilde{\mathbf{R}}_{\theta,t}(\mathcal{G}, \bar{\mathcal{G}}) \right| \leq C_1 \sqrt{C_2 + C_5 \mathbb{E}_{\mathcal{G}_0} q_{t|0}(\mathcal{G}|\mathcal{G}_0) \sum_i \mathcal{L}_{\texttt{CE}}(\texttt{One-Hot}(f_0^i), \hat{f}_0^i)}$$

$$+ C_3 \sqrt{C_4 + C_5 \mathbb{E}_{\mathcal{G}_0} q_{t|0}(\mathcal{G}|\mathcal{G}_0) \sum_{i,j} \mathcal{L}_{\texttt{CE}}(\texttt{One-Hot}(e_0^{(i,j)}), \hat{e}_0^{(i,j)})} \tag{36}$$

$$\overset{(*)}{\leq} \left( C_t + C_t^{\texttt{node}} \mathbb{E}_{\mathcal{G}_0} q_{t|0}(\mathcal{G}|\mathcal{G}_0) \sum_i \mathcal{L}_{\texttt{CE}}(\texttt{One-Hot}(f_0^i), \hat{f}_0^i) \right.$$

$$\left. + C_t^{\texttt{edge}} \mathbb{E}_{\mathcal{G}_0} q_{t|0}(\mathcal{G}|\mathcal{G}_0) \sum_{i,j} \mathcal{L}_{\texttt{CE}}(\texttt{One-Hot}(e_0^{(i,j)}), \hat{e}_0^{(i,j)}) \right)^{1/2} \tag{37}$$

where (*) is based on Cauchy–Schwarz inequality, $C_t = 2C_1^2 C_2 + 2C_3^2 C_4$, $C_t^{\text{node}} = 2C_1^2 C_5$, $C_t^{\text{edge}} = 2C_3^2 C_5$. $\qquad \square$

## C.3 Proof of Lemma 3.5

We clarify that the term "permutation" in this paper refers to the reordering of the node indices, i.e., the first dimension of $\mathbf{F}$ and the first two dimensions of $\mathbf{E}$.

*Proof.* The input of an MPNN layer is $\mathbf{F} = \{\mathbf{r}_i\}_{i=1}^n \in \mathbb{R}^{n \times d}, \mathbf{E} = \{\mathbf{r}_{i,j}\}_{i,j=1}^n \in \mathbb{R}^{n \times n \times d}, \mathbf{y} \in \mathbb{R}^d$, where $d$ is the hidden dimension. The updating formulas of an MPNN layer can be presented as

$$\mathbf{r}^i \leftarrow \texttt{FiLM}\left(\texttt{FiLM}\left(\mathbf{r}^i, \texttt{MLP}\left(\frac{\sum_{j=1}^n \mathbf{r}^{(j,i)}}{n}\right)\right), \mathbf{y}\right), \tag{38}$$

$$\mathbf{r}^{(i,j)} \leftarrow \texttt{FiLM}\left(\texttt{FiLM}(\mathbf{r}^{(i,j)}, \mathbf{r}^i \odot \mathbf{r}^j), \mathbf{y}\right), \tag{39}$$

$$\mathbf{y} \leftarrow \mathbf{y} + \texttt{PNA}\left(\{\mathbf{r}^i\}_{i=1}^n\right) + \texttt{PNA}\left(\{\mathbf{r}^{(i,j)}\}_{i,j=1}^n\right), \tag{40}$$

The permutation $\mathcal{P}$ of the input of an MPNN layer can be presented as $\mathcal{P}\left(\mathbf{F} = \{\mathbf{r}_i\}_{i=1}^n, \mathbf{E} = \{\mathbf{r}_{i,j}\}_{i,j=1}^n, \mathbf{y}\right) = \left(\{\mathbf{r}_{\sigma(i)}\}_{i=1}^n, \{\mathbf{r}_{\sigma(i),\sigma(j)}\}_{i,j=1}^n, \mathbf{y}\right)$ where $\sigma : \{1, \ldots, n\} \mapsto \{1, \ldots, n\}$ is a bijection.

For PNA (Eq. (70)), it includes operations `max`, `min`, `mean`, and `std` which are all permutation-invariant and thus, the PNA module is permutation-invariant. Then,

$$\mathbf{y} + \texttt{PNA}\left(\{\mathbf{r}^i\}_{i=1}^n\right) + \texttt{PNA}\left(\{\mathbf{r}^{(i,j)}\}_{i,j=1}^n\right) = \mathbf{y} + \texttt{PNA}\left(\{\mathbf{r}^{\sigma(i)}\}_{i=1}^n\right) + \texttt{PNA}\left(\{\mathbf{r}^{(\sigma(i),\sigma(j))}\}_{i,j=1}^n\right) \tag{41}$$

Because $\sum_{j=1}^n \mathbf{r}^{(j,i)} = \sum_{j=1}^n \mathbf{r}^{(\sigma(j),\sigma(i))}$, $\mathbf{r}^i \odot \mathbf{r}^j = \mathbf{r}^{\sigma(i)} \odot \mathbf{r}^{\sigma(j)}$, and the FiLM module (Eq. (71)) is not related to the node ordering,

$$\mathbf{r}^{(\sigma(i),\sigma(j))} \leftarrow \texttt{FiLM}\left(\texttt{FiLM}(\mathbf{r}^{(\sigma(i),\sigma(j))}, \mathbf{r}^{\sigma(i)} \odot \mathbf{r}^{\sigma(j)}), \mathbf{y}\right) = \texttt{FiLM}\left(\texttt{FiLM}(\mathbf{r}^{(i,j)}, \mathbf{r}^i \odot \mathbf{r}^j), \mathbf{y}\right) \tag{42}$$

$$\mathbf{r}^{\sigma(i)} \leftarrow \texttt{FiLM}\left(\texttt{FiLM}\left(\mathbf{r}^{\sigma(i)}, \texttt{MLP}\left(\frac{\sum_{j=1}^n \mathbf{r}^{(\sigma(j),\sigma(i))}}{n}\right)\right), \mathbf{y}\right) \tag{43}$$

$$= \texttt{FiLM}\left(\texttt{FiLM}\left(\mathbf{r}^i, \texttt{MLP}\left(\frac{\sum_{j=1}^n \mathbf{r}^{(j,i)}}{n}\right)\right), \mathbf{y}\right) \tag{44}$$

Thus, we proved that

$$\texttt{MPNN}\left(\mathcal{P}(\mathbf{F}, \mathbf{E}, \mathbf{y})\right) = \mathcal{P}\left(\texttt{MPNN}(\mathbf{F}, \mathbf{E}, \mathbf{y})\right) \tag{45}$$

$\qquad \square$

## C.4 Proof of Lemma 3.6

*Proof.* The forward rate matrix (Eq. (3)) is the sum of component-specific forward rate matrices ($\{\mathbf{R}_t^{(i,j)}\}_{i,j \in \mathbb{N}_{\leq n}^+}$ and $\{\mathbf{R}_t^i\}_{i \in \mathbb{N}_{\leq n}^+}$). It is permutation-invariant because the summation is permutation-invariant.

The parametric reverse rate matrix is

$$\tilde{\mathbf{R}}_{\theta,t}(\mathcal{G}, \bar{\mathcal{G}}) = \sum_i \tilde{\mathbf{R}}_{\theta,t}^i(f^i, \bar{f}^i) + \sum_{i,j} \tilde{\mathbf{R}}_{\theta,t}^{(i,j)}(e^{(i,j)}, \bar{e}^{(i,j)}) \tag{46}$$

where $\tilde{\mathbf{R}}_{\theta,t}^i(f^i, \bar{f}^i) = A_t^i \sum_{f_0^i} \frac{q_{t|0}(\bar{f}^i | f_0^i)}{q_{t|0}(f^i | f_0^i)} p_{0|t}^\theta(f_0^i | \mathcal{G}_t)$, $\tilde{\mathbf{R}}_{\theta,t}^{(i,j)}(e^{(i,j)}, \bar{e}^{(i,j)}) = B_t^{(i,j)} \sum_{e_0^{(i,j)}} \frac{q_{t|0}(\bar{e}^{(i,j)} | e_0^{(i,j)})}{q_{t|0}(e^{(i,j)} | e_0^{(i,j)})} p_{0|t}^\theta(e_0^{(i,j)} | \mathcal{G}_t)$. If we present the permutation $\mathcal{P}$ on every node

as a bijection $\sigma : \{1, \ldots, n\} \mapsto \{1, \ldots, n\}$, the term

$$\tilde{\mathbf{R}}_{\theta,t}^i(f^i, \bar{f}^i) = A_t^i \sum_{f_0^i} \frac{q_{t|0}(\bar{f}^i|f_0^i)}{q_{t|0}(f^i|f_0^i)} p_{0|t}^\theta(f_0^i|\mathcal{G}_t) \tag{47}$$

$$= \mathbf{R}_t^i(\bar{f}^i, f^i) \delta_{\bar{\mathcal{G}} \backslash \bar{f}^i, \mathcal{G} \backslash f^i} \sum_{f_0^i} \frac{q_{t|0}(\bar{f}^i|f_0^i)}{q_{t|0}(f^i|f_0^i)} p_{0|t}^\theta(f_0^i|\mathcal{G}_t) \tag{48}$$

$$\overset{(*)}{=} \mathbf{R}_t^{\sigma(i)}(\bar{f}^{\sigma(i)}, f^{\sigma(i)}) \delta_{\mathcal{P}(\bar{\mathcal{G}}) \backslash \bar{f}^{\sigma(i)}, \mathcal{P}(\mathcal{G}) \backslash f^{\sigma(i)}} \sum_{f_0^{\sigma(i)}} \frac{q_{t|0}(\bar{f}^{\sigma(i)}|f_0^{\sigma(i)})}{q_{t|0}(f^{\sigma(i)}|f_0^{\sigma(i)})} p_{0|t}^\theta(f_0^i|\mathcal{G}_t) \tag{49}$$

$$\overset{(**)}{=} \mathbf{R}_t^{\sigma(i)}(\bar{f}^{\sigma(i)}, f^{\sigma(i)}) \delta_{\mathcal{P}(\bar{\mathcal{G}}) \backslash \bar{f}^{\sigma(i)}, \mathcal{P}(\mathcal{G}) \backslash f^{\sigma(i)}} \sum_{f_0^{\sigma(i)}} \frac{q_{t|0}(\bar{f}^{\sigma(i)}|f_0^{\sigma(i)})}{q_{t|0}(f^{\sigma(i)}|f_0^{\sigma(i)})} p_{0|t}^\theta(f_0^{\sigma(i)}|\mathcal{P}(\mathcal{G}_t)) \tag{50}$$

$$= \tilde{\mathbf{R}}_{\theta,t}^{\sigma(i)}(f^{\sigma(i)}, \bar{f}^{\sigma(i)}) \tag{51}$$

where (*) is based on the permutation invariant of the forward process and its rate matrix; (**) is based on the permutation equivariance of the graph-to-graph backbone $p_{0|t}^\theta$.  □

## C.5 Proof of Lemma 3.7

Recall the Kolmogorov forward equation, for $s < t$,

$$\frac{d}{dt} q_{t|s}(\mathbf{x}_t|\mathbf{x}_s) = \sum_{\xi \in \mathcal{X}} q_{t|s}(\xi|\mathbf{x}_s) \mathbf{R}_t(\xi, \mathbf{x}_t). \tag{52}$$

*Proof.* We aim to show that $q_{t|s}(\mathcal{P}(\mathbf{x}_t)|\mathcal{P}(\mathbf{x}_s))$ is a solution of Eq. (52). Because the permutation $\mathcal{P}$ is a bijection, we have

$$\frac{d}{dt} q_{t|s}(\mathcal{P}(\mathbf{x}_t)|\mathcal{P}(\mathbf{x}_s)) \tag{53}$$

$$= \sum_{\xi \in \mathcal{X}} q_{t|s}(\mathcal{P}(\xi)|\mathcal{P}(\mathbf{x}_s)) \mathbf{R}_t(\mathcal{P}(\xi), \mathcal{P}(\mathbf{x}_t)) \tag{54}$$

$$\overset{(*)}{=} \sum_{\xi \in \mathcal{X}} q_{t|s}(\mathcal{P}(\xi)|\mathcal{P}(\mathbf{x}_s)) \mathbf{R}_t(\xi, \mathbf{x}_t) \tag{55}$$

where (*) is because $\mathbf{R}_t$ is permutation-invariant. As Eq. 55 and Eq. 52 share the same rate matrix, and the rate matrix completely determines the CTMC (and its Kolmogorov forward equation) [63], thus, their solutions are the same: $q_{t|s}(\mathbf{x}_t|\mathbf{x}_s) = q_{t|s}(\mathcal{P}(\mathbf{x}_t)|\mathcal{P}(\mathbf{x}_s))$, i.e., the transition probability is permutation-invariant.  □

## C.6 Proof of Theorem 3.8

*Proof.* We start from a simple case where the parametric rate matrix is fixed all the time,

$$p_0^\theta(\mathcal{G}_0) = \sum_{\mathcal{G}_T} q_{0|T}^\theta(\mathcal{G}_0|\mathcal{G}_T) \pi_{\texttt{ref}}(\mathcal{G}_T), \tag{56}$$

where the transition probability is by solving the Kolmogorov forward equation

$$\frac{d}{dt} q_{t|s}^\theta(\mathcal{G}_t|\mathcal{G}_s) = \sum_\xi q_{t|s}^\theta(\xi|\mathcal{G}_s) \tilde{\mathbf{R}}_\theta(\xi, \mathcal{G}_t). \tag{57}$$

Thus, the sampling probability of permuted graph $\mathcal{P}(\mathcal{G}_0)$

$$p_0^\theta(\mathcal{P}(\mathcal{G}_0)) = \sum_{\mathcal{G}_T} q_{0|T}^\theta(\mathcal{P}(\mathcal{G}_0)|\mathcal{P}(\mathcal{G}_T))\pi_{\texttt{ref}}(\mathcal{P}(\mathcal{G}_T)) \tag{58}$$

$$\overset{(*)}{=} \sum_{\mathcal{G}_T} q_{0|T}^\theta(\mathcal{G}_0|\mathcal{G}_T)\pi_{\texttt{ref}}(\mathcal{P}(\mathcal{G}_T)) \tag{59}$$

$$\overset{(**)}{=} \sum_{\mathcal{G}_T} q_{0|T}^\theta(\mathcal{G}_0|\mathcal{G}_T)\pi_{\texttt{ref}}(\mathcal{G}_T) \tag{60}$$

$$= p_0^\theta(\mathcal{G}_0) \tag{61}$$

where (*) is based on Lemma 3.6 and Lemma 3.7, the transition probability of DISCO is permutation-invariant and (**) is from the assumption that the reference distribution $\pi_{\texttt{ref}}(\mathcal{G}_T)$ is permutation-invariant. Thus, we proved that for the simple case, $\tilde{\mathbf{R}}_{\theta,t}$ fixed $\forall t$, the sampling probability is permutation-invariant.

For the practical sampling, as we mentioned in Section 3.4, the $\tau$-leaping algorithm assumes that the time interval $[0, T]$ is divided into various length-$\tau$ intervals $[0, \tau), [\tau, 2\tau), \ldots, [T - \tau, T]$ (here both close sets or open sets work) and assume the reverse rate matrix is fixed as $\tilde{\mathbf{R}}_{\theta,t}$ within every length-$\tau$ interval, such as $(t - \tau, t]$. Thus, the sampling probability can be computed as

$$p_0^\theta(\mathcal{G}_0) = \sum_{\mathcal{G}_T, \mathcal{G}_{T-\tau}, \ldots, \mathcal{G}_\tau} q_{0|\tau}(\mathcal{G}_0|\mathcal{G}_\tau) \ldots q_{T-\tau|T}(\mathcal{G}_{T-\tau}|\mathcal{G}_T)\pi_{\texttt{ref}}(\mathcal{G}_T). \tag{62}$$

The conclusion from the simple case can be generalized to this $\tau$-leaping-based case because all the transition probability $q_{t-\tau|t}(\mathcal{G}_{t-\tau}|\mathcal{G}_t)$ and the reference distribution are permutation-invariant. $\quad\square$

Note that Xu et al. [77] have a similar analysis in their Proposition 1 on a DDPM-based model.

### C.7 Proof of Theorem 3.9

Recall our training objective is

$$\min_\theta T\mathbb{E}_{t\sim\mathcal{U}_{(0,T)}}\mathbb{E}_{\mathcal{G}_0}\mathbb{E}_{q_{t|0}(\mathcal{G}_t|\mathcal{G}_0)}\left[\sum_i \mathcal{L}_{\texttt{CE}}(\texttt{One-Hot}(f_0^i), \hat{f}_0^i) + \sum_{i,j} \mathcal{L}_{\texttt{CE}}(\texttt{One-Hot}(e_0^{(i,j)}), \hat{e}_0^{(i,j)})\right] \tag{63}$$

where $\hat{f}_0^i = [p_{0|t}^\theta(f^i = 1|\mathcal{G}_t), \ldots, p_{0|t}^\theta(f^i = b|\mathcal{G}_t)]^\top \in [0, 1]^b$ and $\hat{e}_0^{(i,j)} = [p_{0|t}^\theta(e^{(i,j)} = 1|\mathcal{G}_t), \ldots, p_{0|t}^\theta(e^{(i,j)} = a + 1|\mathcal{G}_t)]^\top \in [0, 1]^{a+1}$

*Proof.* We follow the notation and present the permutation $\mathcal{P}$ on every node as a bijection $\sigma : \{1, \ldots, n\} \mapsto \{1, \ldots, n\}$. We first analyze the cross-entropy loss on the nodes for a single training graph $\mathcal{G}_0$ and taking expectation $\mathbb{E}_{\mathcal{G}_0}$ keeps the permutation invariance:

$$\mathcal{L}_{\texttt{node}}(\mathcal{G}_0) = T\mathbb{E}_{t\sim\mathcal{U}_{(0,T)}}\mathbb{E}_{q_{t|0}(\mathcal{G}_t|\mathcal{G}_0)}\sum_i \mathcal{L}_{\texttt{CE}}(\texttt{One-Hot}(f_0^i), \hat{f}_0^i) \tag{64}$$

$$= T\mathbb{E}_{t\sim\mathcal{U}_{(0,T)}}\sum_{\mathcal{G}_t} q_{t|0}(\mathcal{G}_t|\mathcal{G}_0)\sum_i \mathcal{L}_{\texttt{CE}}(\texttt{One-Hot}(f_0^i), \hat{f}_0^i) \tag{65}$$

$$\overset{(*)}{=} T\mathbb{E}_{t\sim\mathcal{U}_{(0,T)}}\sum_{\mathcal{G}_t} q_{t|0}(\mathcal{P}(\mathcal{G}_t)|\mathcal{P}(\mathcal{G}_0))\sum_i \mathcal{L}_{\texttt{CE}}(\texttt{One-Hot}(f_0^i), \hat{f}_0^i) \tag{66}$$

$$\overset{(**)}{=} T\mathbb{E}_{t\sim\mathcal{U}_{(0,T)}}\sum_{\mathcal{G}_t} q_{t|0}(\mathcal{P}(\mathcal{G}_t)|\mathcal{P}(\mathcal{G}_0))\sum_i \mathcal{L}_{\texttt{CE}}(\texttt{One-Hot}(f_0^{\sigma(i)}), \hat{f}_0^{\sigma(i)}) \tag{67}$$

$$= \mathcal{L}_{\texttt{node}}(\mathcal{P}(\mathcal{G}_0)) \tag{68}$$

where (*) is from the permutation invariance of the forward process and (**) is from the permutation equivariance of the graph-to-graph backbone and the permutation invariance of the cross-entropy

loss. A similar result can be analyzed on the cross-entropy loss on the edges

$$\mathcal{L}_{\text{edge}}(\mathcal{G}_0) = T\mathbb{E}_{t \sim \mathcal{U}_{(0,T)}} \mathbb{E}_{q_{t|0}(\mathcal{G}_t|\mathcal{G}_0)} \sum_{i,j} \mathcal{L}_{\text{CE}}(\text{One-Hot}(e_0^{(i,j)}), \hat{e}_0^{(i,j)}) = \mathcal{L}_{\text{edge}}(\mathcal{P}(\mathcal{G}_0)) \quad (69)$$

and we omit the proof here for brevity. $\qquad\square$

## D  Sampling Algorithm

A Step-by-step procedure about the $\tau$-leaping graph generation is presented in Algorithm 2.

---

**Algorithm 2** $\tau$-Leaping Graph Generation

---

1: $t \leftarrow T$
2: $\mathcal{G}_t = (\{e^{(i,j)}\}_{i,j \in \mathbb{N}^+_{\leq n}}, \{f^i\}_{i \in \mathbb{N}^+_{\leq n}}) \leftarrow \pi_{\texttt{ref}}(\mathcal{G})$
3: **while** $t > 0$ **do**
4:     **for** $i = 1, \ldots, n$ **do**
5:         **for** $s = 1, \ldots, b$ **do**
6:             $\tilde{\mathbf{R}}^i_{\theta,t}(f^i, s) = \mathbf{R}^i_t(s, f^i) \sum_{f_0^i} \frac{q_{t|0}(s|f_0^i)}{q_{t|0}(f^i|f_0^i)} p_\theta(f^i|\mathcal{G}_t, t)$
7:             $J_{f^i,s} \leftarrow \text{Poisson}(\tau\mathbf{R}^i_t(s, f^i))$                ▷ # of transition for every node
8:         **end for**
9:     **end for**
10:     **for** $i, j = 1, \ldots, n$ **do**
11:         **for** $s = 1, \ldots, a$ **do**
12:             $\tilde{\mathbf{R}}^{(i,j)}_{\theta,t}(e^{(i,j)}, s) = \mathbf{R}^{(i,j)}_t(s, e^{(i,j)}) \sum_{e_0^{(i,j)}} \frac{q_{t|0}(s|e_0^{(i,j)})}{q_{t|0}(e^{(i,j)}|e_0^{(i,j)})} p_\theta(e^{(i,j)}|\mathcal{G}_t, t)$
13:             $J_{e^{(i,j)},s} \leftarrow \text{Poisson}(\tau\mathbf{R}^{(i,j)}_t(s, e^{(i,j)}))$        ▷ # of transition for every edge
14:         **end for**
15:     **end for**
16:     **for** $i = 1, \ldots, n$ **do**
17:         **if** $\sum_{s=1}^b J_{f^i,s} > 1$ or $\sum_{s=1}^b J_{f^i,s} = 0$ **then**
18:             $f^i \leftarrow f^i$                               ▷ stay the same
19:         **else**
20:             $s^* = \arg\max_s\{J_{f^i,s}\}_{s=1}^b$
21:             $f^i \leftarrow s^*$                              ▷ update node
22:         **end if**
23:     **end for**
24:     **for** $i, j = 1, \ldots, n$ **do**
25:         **if** $\sum_{s=1}^a J_{e^{(i,j)},s} > 1$ or $\sum_{s=1}^a J_{e^{(i,j)},s} = 0$ **then**
26:             $e^{(i,j)} \leftarrow e^{(i,j)}$                       ▷ stay the same
27:         **else**
28:             $s^* = \arg\max_s\{J_{e^{(i,j)},s}\}_{s=1}^b$
29:             $e^{(i,j)} \leftarrow s^*$                        ▷ update edge
30:         **end if**
31:     **end for**
32:     $t \leftarrow t - \tau$
33: **end while**

---

## E  Auxiliary Features, PNA and FiLM Modules

For learning a better graph-to-graph mapping $p_{0|t}^\theta(\mathcal{G}_0|\mathcal{G}_t)$, artificially augmenting the node-level features and graph-level features is proved effective to enhance the expressiveness of graph learning models [81, 73]. For this setting, we keep consistent with the state-of-the-art model, DiGress [73], and extract the following three sets of auxiliary features. Note that the following features are extracted on the noised graph $\mathcal{G}_t$.

We binarize the edge tensor $\mathbf{E}$ into an adjacency matrix $\mathbf{A} \in \{0,1\}^{n \times n}$ whose 1 entries denote the corresponding node pair is connected by any type of edge.

**Motif features.** The number of length-$3/4/5$ cycles every node is included in is counted as the topological node-level features; also, the total number of length-$3/4/5/6$ cycles is the topological graph-level features.

**Spectral features.** The graph Laplacian is decomposed. The number of connected components and the first 5 non-zero eigenvalues are selected as the spectral graph-level features. An estimated indicator of whether a node is included in the largest connected component and the first 2 eigenvectors of the non-zero eigenvalues are selected as the spectral node-level features.

**Molecule features.** On molecule datasets, the valency of each atom is selected as the node-level feature, and the total weight of the whole molecule is selected as the graph-level feature.

The above node-level features and graph-level features are concatenated together as the auxiliary node-level features $\mathbf{F}_{\text{aux}}$ and graph-level features $\mathbf{y}$. An important property is that the above node-level features are permutation-equivariant and the above graph-level features are permutation-invariant, whose proof is straightforward so we omit it here.

Next, two important modules used in the MPNN backbone: PNA and FiLM are detailed.

**PNA module.** The PNA module [12] is implemented as follows,

$$\texttt{PNA}(\{\mathbf{x}_i\}_{i=1}^n) = \texttt{MLP}(\texttt{min}(\{\mathbf{x}_i\}_{i=1}^n) \oplus \texttt{max}(\{\mathbf{x}_i\}_{i=1}^n) \oplus \texttt{mean}(\{\mathbf{x}_i\}_{i=1}^n) \oplus \texttt{std}(\{\mathbf{x}_i\}_{i=1}^n)) \quad (70)$$

where $\oplus$ is the concatenation operator, $\mathbf{x}_i \in \mathbb{R}^d$; $\texttt{min}$, $\texttt{max}$, $\texttt{mean}$, and $\texttt{std}$ are coordinate-wise, e.g., $\texttt{min}(\{\mathbf{x}_i\}_{i=1}^n) \in \mathbb{R}^d$.

**FiLM module.** FiLM [57] is implemented as follows,

$$\texttt{FiLM}(\mathbf{x}_i, \mathbf{x}_j) = \texttt{Linear}(\mathbf{x}_i) + \texttt{Linear}(\mathbf{x}_i) \odot \mathbf{x}_j + \mathbf{x}_j \quad (71)$$

where $\texttt{Linear}$ is a single fully-connected layer without activation function and $\odot$ is the Hadamard product.

# F  Supplementary Details about Experiments

## F.1  Hardware and Software

We implement DISCO in PyTorch[5] and PyTorch-geometric[6]. All the efficiency study results are from one NVIDIA Tesla V100 SXM2-32GB GPU on a server with 96 Intel(R) Xeon(R) Gold 6240R CPU @ 2.40GHz processors and 1.5T RAM. The training on QM9 and Community can be finished in 2 hours. For the training on SBM, Planar, it can be finished within 48 hours to get decent validity. The training on MOSES and GuacaMol can be finished within 96 hours.

## F.2  Dataset Setup

### F.2.1  Dataset Statistics

The statistics about all the datasets used in this paper are presented in Table 7, where $a$ is the number of edge types, $b$ is the number of node types, $|\mathbf{E}|$ is the number of edges and $|\mathbf{F}|$ is the number of nodes.

### F.2.2  Detailed Settings on Plain Graph Datasets

**Dataset Split.** We follow the settings of SPECTRE [51] and DiGress [73] to split the SBM, Planar [51], and Community [82] datasets into $64/16/20\%$ for training/validation/test set.

---

[5]https://pytorch.org
[6]https://pytorch-geometric.readthedocs.io/en/latest

Table 7: Dataset statistics.

| Name | # Graphs | Split | $a$ | $b$ | Avg. $|\mathbf{E}|$ | Max $|\mathbf{E}|$ | Avg. $|\mathbf{F}|$ | Max $|\mathbf{F}|$ |
|------|----------|-------|-----|-----|------|------|------|------|
| SBM | 200 | 128/32/40 | 1 | 1 | 1000.8 | 2258 | 104.0 | 187 |
| Planar | 200 | 128/32/40 | 1 | 1 | 355.7 | 362 | 64.0 | 64 |
| Community | 100 | 64/16/20 | 1 | 1 | 74.0 | 122 | 15.7 | 20 |
| QM9 | 130831 | 97734/20042/13055 | 4 | 4 | 18.9 | 28 | 8.8 | 9 |
| MOSES | 1733214 | 1419512/156176/157526 | 4 | 8 | 46.3 | 62 | 21.6 | 27 |
| GuacaMol | 1398213 | 1118633/69926/209654 | 4 | 12 | 60.4 | 176 | 27.8 | 88 |

**Metrics.** The Maximum Mean Discrepancy (MMD) [82] measures the discrepancy between two sets of distributions. The relative squared MMD [73]is defined as follows

$$score = \frac{\text{MMD}^2(\{\mathcal{G}\}_{\texttt{gen}}||\{\mathcal{G}\}_{\texttt{test}})}{\text{MMD}^2(\{\mathcal{G}\}_{\texttt{train}}||\{\mathcal{G}\}_{\texttt{test}})}, \tag{72}$$

where $(\{\mathcal{G}\}_{\texttt{gen}}$, $(\{\mathcal{G}\}_{\texttt{train}}$, and $(\{\mathcal{G}\}_{\texttt{test}}$ are the sets of generated graphs, training graphs, and test graphs, respectively. We report the above relative squared MMD for degree distributions (Deg.), clustering coefficient distributions (Clus.), and average orbit counts (Orb.) statistics (the number of occurrences of all substructures with 4 nodes). In addition, the Uniqueness, Novelty, and Validity are chosen. Uniqueness reports the fraction of the generated nonisomorphic graphs; Novelty reports the fraction of the generated graphs not isomorphic with any graph from the training set; Validity checks the fraction of the generated graphs following some specific rules. For the SBM dataset, we follow the validity check from [51] whose core idea is to check whether real SBM graphs are statistically indistinguishable from the generated graphs; for the Planar dataset, we check whether the generated graphs are connected and are indeed planar graphs. Because the Community dataset does not have the Validity metric, we only report the Uniqueness, Novelty, and Validity results on the SBM and Planar datasets.

We report mean±std in 5 runs.

**Baseline methods.** GraphRNN [82], GRAN [42], GG-GAN [37], MolGAN [9], SPECTRE [51], EDP-GNN [56], GraphGDP [26], DiscDDPM [22], EDGE [10], ConGress [73], DiGress [73] are chosen.

### F.2.3 Detailed Settings on Molecule Graph Datasets

**Dataset Split.** We follow the split of QM9 from DiGress [73] and follow the split of MOSES [58] and GuacaMol [6] according to their benchmark settings. Their statistics are presented in Table 7.

**Metrics.** For QM9, Uniqueness, Novelty, and Validity are chosen as metrics. The first two are the same as introduced in Section F.2.2. The Validity is computed by building a molecule with RdKit [7] and checking if we can obtain a valid SMILES string from it.

For MOSES, the chosen metrics include Uniqueness, Novelty, Validity, Filters, Fréchet ChemNet Distance (FCD), Similarity to a nearest neighbor (SNN), and Scaffold similarity (Scaf), which is consistent with DiGress [73]. The official evaluation code [8] is used to report the performance.

For GuacaMol, the chosen metrics include Uniqueness, Novelty, Validity, KL Divergence, and Fréchet ChemNet Distance (FCD), which is consistent with DiGress [73]. The official evaluation code [9] is used to report the performance.

We report mean±std in 5 runs except MOSES and GuacaMol, whose computations are too expensive to repeat multiple times.

**Baseline methods.** CharacterVAE [20], GrammarVAE [38], GraphVAE [66], GT-VAE [55], Set2GraphVAE [72], GG-GAN [37], MolGAN [9], SPECTRE [51], GraphNVP [50], GDSS [32],

---

[7]https://www.rdkit.org/
[8]https://github.com/molecularsets/moses
[9]https://github.com/BenevolentAI/guacamol

Table 8: Generation performance (mean±std) on the Community dataset.

| Model | Deg.↓ | Clus.↓ | Orb.↓ |
|---|---|---|---|
| GraphRNN [82] | 4.0 | 1.7 | 4.0 |
| GRAN [42] | 3.0 | 1.6 | 1.0 |
| EDP-GNN [56] | 2.5 | 2.0 | 3.0 |
| GraphGDP [26] | 2.0 | 1.1 | - |
| DiscDDPM [22] | 1.2 | **0.9** | 1.5 |
| EDGE [10] | 1.0 | 1.0 | 2.0 |
| GG-GAN [37] | 4.0 | 3.1 | 8.0 |
| MolGAN [9] | 3.0 | 1.9 | 1.0 |
| SPECTRE [51] | 0.5 | 2.7 | 2.0 |
| DiGress [73] | 1.0 | **0.9** | 1.0 |
| DISCO-MPNN | $1.4_{\pm 0.5}$ | $\mathbf{0.9}_{\pm \mathbf{0.2}}$ | $\mathbf{0.9}_{\pm \mathbf{0.3}}$ |
| DISCO-GT | $\mathbf{0.9}_{\pm \mathbf{0.2}}$ | $\mathbf{0.9}_{\pm \mathbf{0.3}}$ | $1.1_{\pm 0.4}$ |

EDGE [10], ConGress [73], DiGress [73], GRAPHARM [36],VAE [21], JT-VAE [29], GraphIN-VENT [53], LSTM [64], NAGVAE [40], and MCTS [28] are chosen.

### F.3 Hyperparameter Settings

**Forward Diffusion Settings.** As we introduced in Proposition 3.2, we tried two sets of rate matrices for the node and edge forward diffusion, so that the converged distribution is either uniform or marginal distribution. We found the marginal distribution leads to better results than the uniform distribution. Thus, the reference distribution is the marginal distribution for all the main results, except Tables 6 and 9. The performance comparison between the marginal diffusion and uniform diffusion is presented in the ablation study in Sections 4.4 andF.5. The $\beta(t)$ controls how fast the forward process converges to the reference distribution, which is set as $\beta(t) = \alpha \gamma^t log(\gamma)$, which is consistent with many existing works [23, 70, 8]. In our implementation, we assume the converged time $T = 1$ and for the forward diffusion hyperparameters $(\alpha, \gamma)$ we tried two sets: $(1.0, 5.0)$ and $(0.8, 2.0)$ where the former one can ensure at $T = 1$ the distribution is very close to the reference distribution, and the latter one does not fully corrupt the raw data distribution so the graph-to-graph model $p_{0|t}^\theta$ is easier to train.

**Reverse Sampling Settings.** The number of sampling steps is determined by $\tau$, which is $\mathtt{round}(\frac{1}{\tau})$ if we set the converged time $T = 1$. We select the number of sampling steps from $\{50, 100, 500\}$, which is much smaller the number of sampling steps of DiGress [73] from $\{500, 1000\}$. For the number of nodes $n$ in every generated graph, we compute a graph size distribution of the training set by counting the number of graphs for different sizes (and normalize the counting to sum it up to 1). Then, we will sample the number of nodes from this graph size distribution for graph generation.

**Neural Network Settings.** For DISCO-GT, the parametric graph-to-graph model $p_{0|t}^\theta$ is graph transformer (GT). We use the exactly same GT architecture as DiGress [73] and adopt their recommended configurations [10]. The reason is that this architecture is not our contribution, and setting the graph-to-graph model $p_{0|t}^\theta$ same can ensure a fair comparison between the discrete-time graph diffusion framework (from DiGress) and the continuous-time graph diffusion framework (from this work). For DISCO-MPNN, we search the number of MPNN layers from $\{3, 5, 8\}$, set all the hidden dimensions the same, and search it from $\{256, 512\}$. For both variants, the dropout is set as $0.1$, the learning rate is set as $2e^{-4}$, and the weight decay is set as $0$.

### F.4 Additional Results on Community

Additional Community plain graph dataset results are in Table 8. Our observation is consistent with the main content: both variants of DISCO are on par with, or even better than the SOTA general graph diffusion generative model, DiGress.

---

[10]https://github.com/cvignac/DiGress/tree/main/configs/experiment

Table 9: Ablation study (mean±std%) with MPNN backbone. V., U., and N. mean Valid, Unique, and Novel.

| Ref. Dist. | Steps | Valid ↑ | V.U. ↑ | V.U.N. ↑ |
|---|---|---|---|---|
| Marginal | 500 | $98.9_{\pm0.7}$ | $98.7_{\pm0.5}$ | $68.7_{\pm0.2}$ |
| | 100 | $98.4_{\pm1.1}$ | $98.0_{\pm1.0}$ | $69.1_{\pm0.6}$ |
| | 30 | $97.7_{\pm1.2}$ | $97.5_{\pm0.8}$ | $70.4_{\pm1.1}$ |
| | 10 | $92.3_{\pm1.9}$ | $91.9_{\pm2.2}$ | $66.4_{\pm1.7}$ |
| | 5 | $88.8_{\pm3.3}$ | $87.1_{\pm2.8}$ | $67.3_{\pm2.9}$ |
| | 1 | $64.4_{\pm2.7}$ | $63.2_{\pm1.9}$ | $55.8_{\pm1.4}$ |
| Uniform | 500 | $93.5_{\pm1.7}$ | $93.2_{\pm1.1}$ | $64.9_{\pm1.0}$ |
| | 100 | $93.1_{\pm2.1}$ | $92.6_{\pm1.7}$ | $66.2_{\pm1.9}$ |
| | 30 | $87.1_{\pm1.8}$ | $86.8_{\pm1.1}$ | $64.0_{\pm1.0}$ |
| | 10 | $83.7_{\pm3.2}$ | $81.9_{\pm2.1}$ | $61.3_{\pm2.0}$ |
| | 5 | $81.5_{\pm2.9}$ | $75.4_{\pm3.4}$ | $64.6_{\pm2.3}$ |
| | 1 | $71.3_{\pm2.3}$ | $42.2_{\pm4.0}$ | $36.9_{\pm3.2}$ |

## F.5 Additional Ablation Study

Table 9 shows the ablation study of DISCO-MPNN on QM9 dataset. Our observations are consistent with the main content: (1) generally, the fewer sampling steps, the lower the generation quality but method's performance is robust in terms of the decreasing of sampling steps; (2) the marginal reference distribution is better than the uniform distribution, consistent with the observation from DiGress [73].

## F.6 Convergence Study

Figure 3 shows the training loss of DISCO-GT and DISCO-MPNN on four datasets, whose X-axis is the number of iterations (i.e., the number of epochs × the number of training samples / batch size). We found that overall the training losses converge smoothly on 4 datasets.

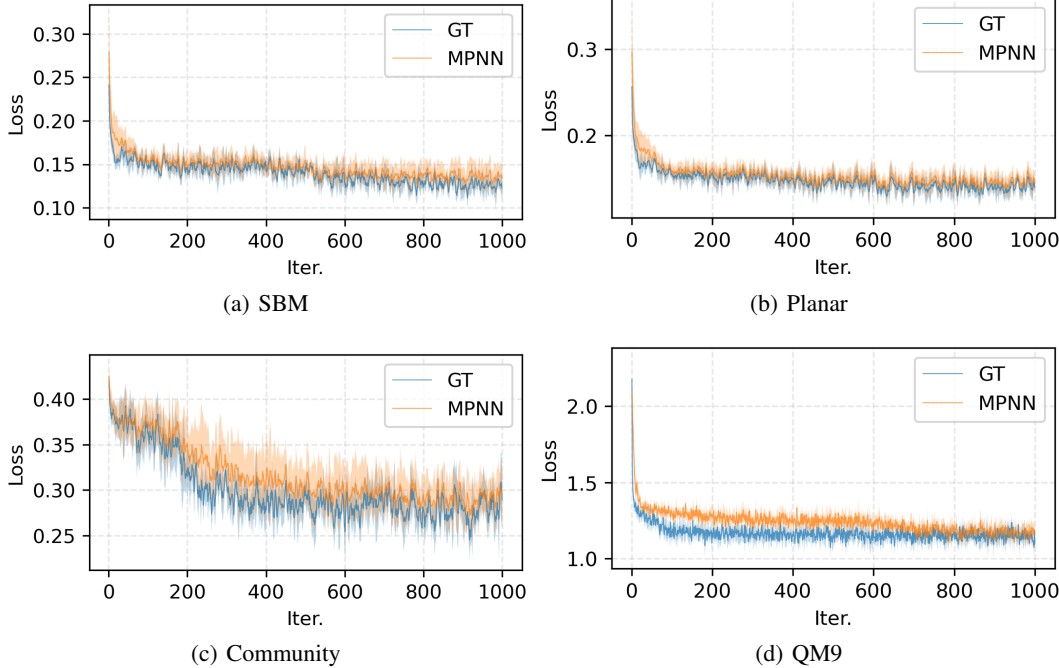

(a) SBM    (b) Planar

(c) Community    (d) QM9

Figure 3: Training loss of DISCO on different datasets and backbone models.

### F.7 Visualization

The generated graphs on the SBM and Planar datasets are presented in Figure 4. We clarify that the generated planar graphs are *selected to be valid* because, as Table 1 shows, not all the generated graphs are valid planar graphs, but the planar layout can only visualize valid planar graphs in our setting [11]. The generated SBM graphs are not selected; even if a part of them cannot pass the strict SBM statistic test (introduced in Section F.2.2 - Metrics), most, if not all, of them still form $2-5$ densely connected clusters.

The generation trajectory of SBM graphs is presented in Figure 5 which demonstrates the reverse denoising process visually.

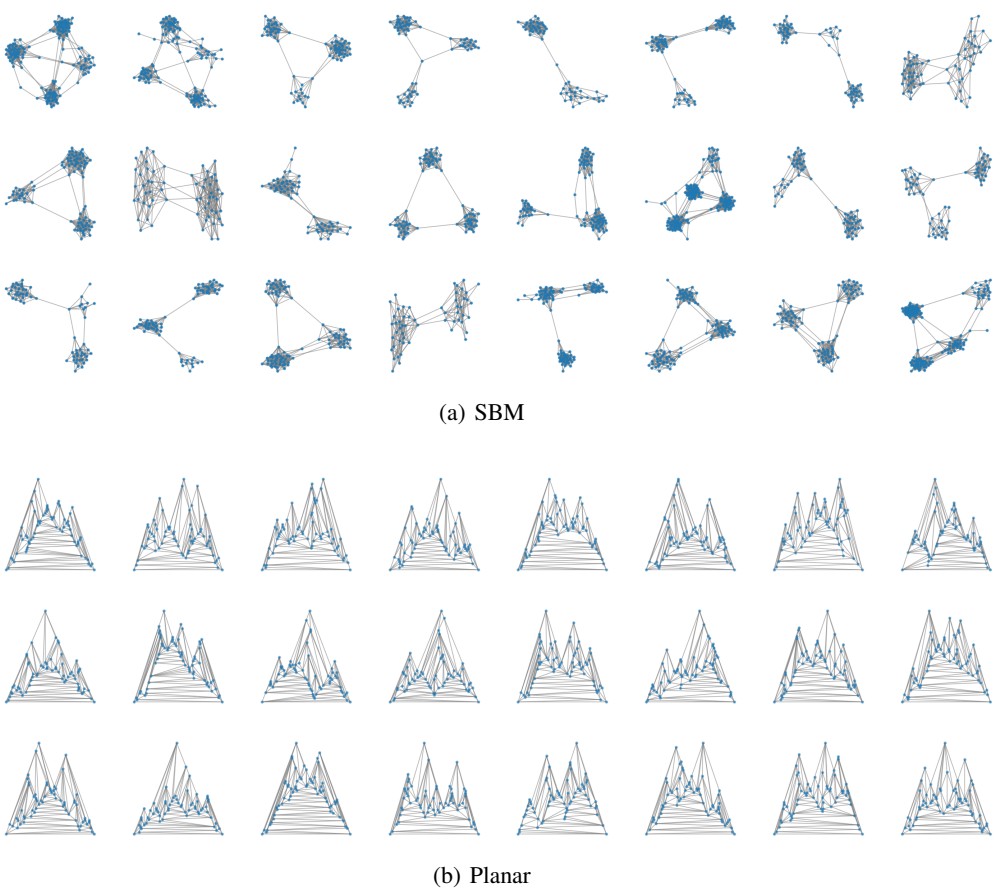

(a) SBM

(b) Planar

Figure 4: Generated graphs.

## G Limitation and Future Work

In this paper, we study the generation of graphs with categorical node and edge types. The current model DISCO cannot be applied to generate graphs with multiple node/edge features (e.g., multiplex networks) and this is an important future work to study. Also, we view the absence of edge as a special type of edge, which forms a complete graph and promotes the expressiveness of our MPNN backbone model. However, it will lead to quadratic complexity concerning the number of nodes. For our current dataset (e.g. graphs with $< 1000$ nodes) the complexity is still acceptable but for future studies on generating *large* graphs, we aim to design more efficient diffusion generative models.

---

[11]`https://networkx.org/documentation/stable/reference/generated/networkx.drawing.layout.planar_layout.html`

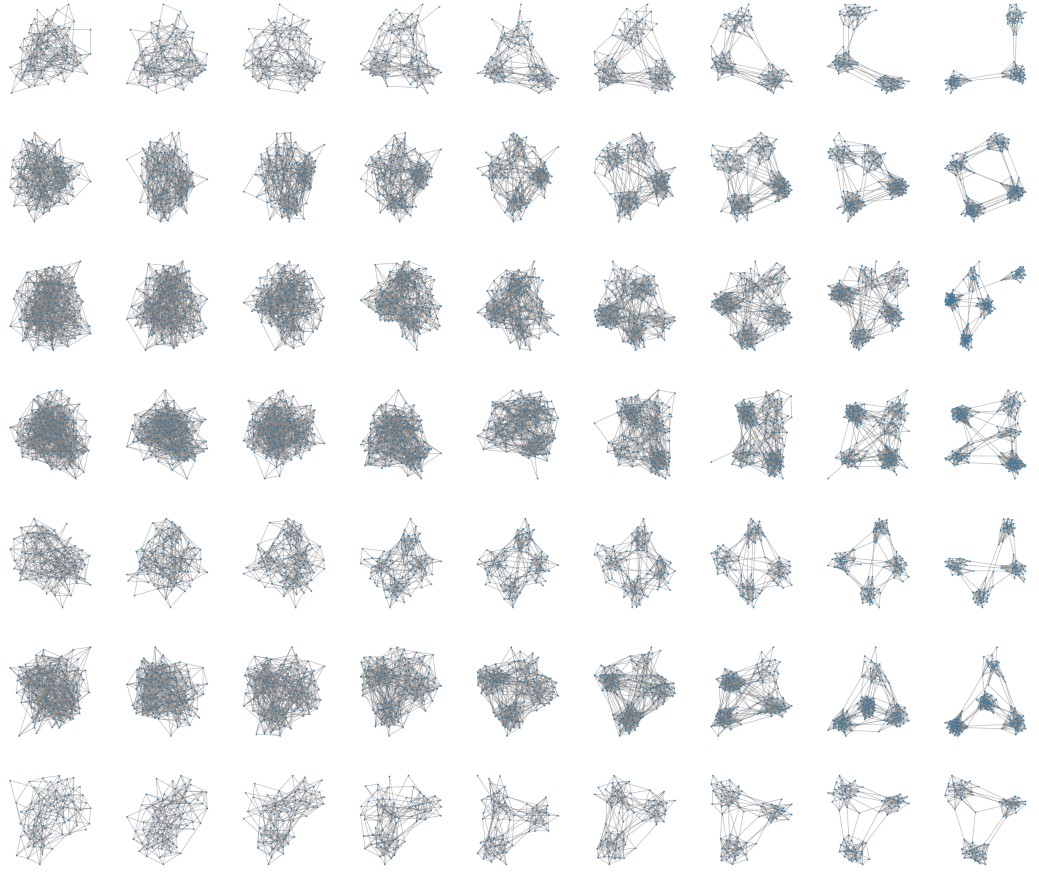

Figure 5: Generation trajectory of SBM graphs with different sizes. Every row is the generation trajectory of one graph from time $t = T$ (left) to $t = 0$ (right) with equal time intervals.

## H    Broader Impact

This paper presents work whose goal is to advance the field of Machine Learning. There are many potential societal consequences of our work, none of which we feel must be specifically highlighted here.

