# OpenReview forum: "Discrete-state Continuous-time Diffusion for Graph Generation"
_NeurIPS.cc/2024/Conference — NeurIPS 2024 poster_

### Official Review · Reviewer_9BTR · 2024-06-23

**Soundness:** 3
**Presentation:** 3
**Contribution:** 3
**Rating:** 6
**Confidence:** 4

**Summary:**

This paper is an extension of current graph diffusion generation to discrete-state and continuous-time version, and proposes DisCo model. The authors also prove permutation-equivariance/invariance of the proposed diffusion framework, sampling and training loss.

**Strengths:**

1. Continuous-time diffusion is a generlization of different strategies, e.g. DDPM and NCSN, so it is more powerful and flexible. Discrete state is also suitable for graph data becaused of discrete edge connection and node classes. This paper integrates these two advantages into a framework, and its good performance is expected.

2. The wrting quality and organization is good and easy to follow.

**Weaknesses:**

The selected baselines are not enough. Only DiGress is compared as graph diffusion generation, but other generation methods based on diffusion have been developed. I only list two relevant baselines  [1-2], but I convince that there are more relevent baselines.

[1] Chen, X., He, J., Han, X., & Liu, L. P. (2023). Efficient and degree-guided graph generation via discrete diffusion modeling. arXiv preprint arXiv:2305.04111.

[2] Fu, X., Gao, Y., Wei, Y., Sun, Q., Peng, H., Li, J., & Li, X. (2024). Hyperbolic Geometric Latent Diffusion Model for Graph Generation. arXiv preprint arXiv:2405.03188.

**Questions:**

As shown in table 3, DisCo-MPNN is much more efficient than DisCo-GT. But from eq. (7) and (8), DisCo-MPNN also conducts aggregation over the whole graph, so its complexity is the same as graph transformer. How to explain this point?

**Limitations:**

I recommend the authors supplement more relevant baselines to enrich the experiments.

---

> ### Author Rebuttal · Authors · 2024-08-05
>
> Dear reviewer 9BTR, we appreciate your tremendous time and effort in reviewing our paper and are glad that you recognize the proposed method's reasonability and writing quality. Our point-to-point responses to your questions are as follows.
>
> > **Q1.** The selected baselines are not enough. Only DiGress is compared as graph diffusion generation, but other generation methods based on diffusion have been developed. I only list two relevant baselines, but I convince that there are more relevent baselines.
>
> **A1.** Thanks for this suggestion.  First, we would like to explain that in our submitted version, we basically select the same baseline methods as Digress (a SOTA discrete graph diffusion generative model) because our goal is to show the proposed method can have **competitive performance against Digress [1]** but with **much better efficiency**; in other words, our goal is **not to pursue a higher generation quality, but a better accuracy-efficiency trade-off**.
>
> Meanwhile, following reviewer's suggestion, we have included more baseline methods' reported performance as follows:
>
> 1. We include results of Digress and our method DisCo from our submitted paper for comparison.
>
> 2. Newly added baselines are commonly tested on different datasets or metrics, so we only **report their performance on the shared datasets and metrics** in our paper during the rebuttal phase. We will include more baselines, datasets, and metrics in the revised version.
>
> 3. We report more results on the Community dataset in Table 1 below, whose metrics are MMD ratios $\textrm{MMD}^2(\{\mathcal{G}\}\_{\texttt{gen}}||\{\mathcal{G}\}\_{\texttt{test}})/\textrm{MMD}^2(\{\mathcal{G}\}\_{\texttt{train}}||\{\mathcal{G}\}\_{\texttt{test}})$ (more details about this metric are in Appendix Section F.2.2) which shows that **DisCo has comparable performance against many SOTA models**. Also, we would like to note that intuitively, **the MMD ratios should not be much lower than 1**, because the distribution of the training set is the optimization objective of our generative models. Interestingly, HypDiff reports performance as low as 0.1, which appears to be significantly lower than other models. We will investigate this further by examining their evaluation and attempting to reproduce their results after this rebuttal phase.
>
> 4. We report more results on the QM9 dataset in Table 2 below, which **also shows that DisCo performs comparably** against SOTA models.
>
> | Model | Deg.$\downarrow$ | Clus.$\downarrow$ | Orb.$\downarrow$ |
> |---|---|---|---|
> |     GNF [2]    | 1.0 | 2.9 | 17.0 |
> | EDP-GNN [3] | 2.5 | 2.0 | 3.0 |
> | GraphGDP [4] | 2.0 | 1.1 | - |
> | GDSS [5] | 2.3 | 1.2 | 0.7 |
> |     DiscDDPM [6]   | 1.2 | 0.9 | 1.5 |
> | EDGE [7] | 1.0 | 1.0 | 2.0 |
> | HypDiff [8] | 0.1 | 0.1 | - |
> | DiGress [1]  | 1.0 | 0.9 | 1.0 |
> | DisCo-GT (Ours) | 0.9$\pm$0.2 | 0.9$\pm$0.3 | 1.1$\pm$0.4 |
> | DisCo-MPNN (Ours) | 1.4$\pm$0.5 | 0.9$\pm$0.2 | 0.9$\pm$0.3 |
> Table 1. More results from baselines on the **Community** Dataset
>
>
> | Model | Valid $\uparrow$ | V.U. $\uparrow$ | V.U.N. $\uparrow$ |
> |---|---|---|---|
> | GDSS [5] | 95.7 | 94.3 | - |
> | EDGE [7] | 99.1 | 99.1 | - |
> | ConGress [1] | 98.9 | 95.7 | 38.3 |
> | DiGress [1] | 99.0 | 95.2 | 31.8 |
> | DisCo-GT (Ours) | 98.0$\pm$0.7 | 97.6$\pm$0.5 | 68.6$\pm$0.8 |
> | DisCo-MPNN (Ours) | 92.8$\pm$2.1 | 91.2$\pm$1.5 | 78.0$\pm$1.1 |
> Table 2. More results from baselines on the **QM9** Dataset
>
> ---
>
> > **Q2.** As shown in table 3, DisCo-MPNN is much more efficient than DisCo-GT. But from eq. (7) and (8), DisCo-MPNN also conducts aggregation over the whole graph, so its complexity is the same as graph transformer. How to explain this point?
>
> **A2.** This is a good question. You are right that the MPNN's aggregation is over the whole graph, as we mentioned between lines 210 and 214 and that is the reason why we believe MPNN can also achieve nice effectiveness. The key architecture difference between GT (from Digress [1]) and our MPNN is that we remove the **multi-head outer product-based self-attention layers** (lines 236-237), as we empirically found their contribution to the generation quality is minor, but are computationally very expensive. In other words, our model can be viewed with **only 1 head without computing the attention**, which leads to the speedup and the decrease of \# of parameters.
>
> P.S., the multi-head outer product-based self-attention layers from Digress can be found in Figure 6 in [1], and their implementation is between lines 182-184 in https://github.com/cvignac/DiGress/blob/main/src/models/transformer_model.py.
>
>
> ```
> Reference:
>
> [1] Vignac, Clement, et al. "DiGress: Discrete Denoising diffusion for graph generation." ICLR 2023.
>
> [2] Liu, Jenny, et al. "Graph normalizing flows." NeurIPS 2019.
>
> [3] Niu, Chenhao, et al. "Permutation invariant graph generation via score-based generative modeling." AISTATS 2020.
>
> [4] Huang, Han, et al. "Graphgdp: Generative diffusion processes for permutation invariant graph generation." ICDM 2022.
>
> [5] Jo, Jaehyeong, Seul Lee, and Sung Ju Hwang. "Score-based generative modeling of graphs via the system of stochastic differential equations." ICML 2022.
>
> [6] Haefeli, Kilian Konstantin, et al. "Diffusion Models for Graphs Benefit From Discrete State Spaces." NeurIPS 2022 GLFrontiers Workshop.
>
> [7] Chen, Xiaohui, et al. "Efficient and Degree-Guided Graph Generation via Discrete Diffusion Modeling." ICML 2023.
>
> [8] Fu, Xingcheng, et al. "Hyperbolic Geometric Latent Diffusion Model for Graph Generation." ICML 2024.
> ```

---

> > ### Comment · Reviewer_9BTR · 2024-08-10
> > **Response to the rebuttal**
> >
> > Thank the authors for more experiments, and further explanation. I choose to keep my positive score.

---

> > > ### Author Response · Authors · 2024-08-11
> > >
> > > We sincerely appreciate your thoughtful review and are glad to have the opportunity to clarify your questions. Thank you for your recognition of our work!

---

### Official Review · Reviewer_XoYX · 2024-07-07

**Soundness:** 3
**Presentation:** 3
**Contribution:** 2
**Rating:** 5
**Confidence:** 3

**Summary:**

The paper presents DISCO (Discrete-State Continuous-Time Diffusion), a novel framework for graph generation. DISCO formulates the graph diffusion process in a discrete-state continuous-time setting, preserving the discrete nature of graph data while allowing for flexible sampling trade-offs between quality and efficiency. The model integrates concepts from continuous-time Markov Chains (CTMC) and employs a graph-to-graph neural network backbone for efficient graph generation.

**Strengths:**

- the paper is well written. most sections are easy to follow

- the proposed research question is important (efficiency is indeed a limitation of current discrete graph generation model)

- the proposed method is technically sound

**Weaknesses:**

> w1: the proposed seems to be a rather straightforward derivation from the existing tools (replacing discrete markov chain with continuous time)

> w2: no implementation available

> w3:  there is a disconnection between the proposed motivation (sampling motivation) and the (theoretical) discussion of the proposed method (see questions below)

> w4: most theoretical results seem to be a standard derivation from existing literature

**Questions:**

> Q1: the motivation for making the original discrete transition process continuous is to improve the sampling efficiency. Has this been achieved in the proposed method? If so, what is the improvement and where exactly does this improvement come from?

**Limitations:**

N.A

---

> ### Author Rebuttal · Authors · 2024-08-05
>
> Dear reviewer XoYX, we appreciate your tremendous time and effort in reviewing our paper and are glad that you recognize its presentation, technical soundness, and importance to this research problem. Our point-to-point responses to your questions are as follows.
>
> > **Q1.** The proposed method seems to be a rather straightforward derivation from the existing tools (replacing discrete markov chain with continuous time). Most theoretical results seem to be a standard derivation from existing literature.
>
> **A1.**  Although there have been prior works (e.g., [1]) on CTMC-based diffusion models, we make the following new contributions:
>
> 1. Existing CTMC-based diffusion models (e.g., [1]) are not permutation-equivariant, thus not suitable for graph generation. We propose a permutation-equivariant version of the CTMC formulation to address this issue. We remark that it is **a highly nontrivial modification of existing discrete-time graph diffusion models**. This is because discrete-time diffusion has a simpler reverse process, but the reverse process of CTMC is different and much more complicated.
>
> 2. We derive a CE-based loss function different from the existing ELBO-based loss for CTMC-based diffusion [1]. The CE-based loss has been widely used in discrete diffusion [2-3] but still lacks a theoretical foundation. To bridge this gap, we prove our Theorem 3.3, showing that the CE-based loss is an upper bound of the estimation error of reverse rate matrices. To our best knowledge, **such analysis does not exist in existing literature.**
>
> 3. The theoretical analysis in Section 3.6 shows the proposed graph generative model's permutation equivariance and invariance. Our analysis is **not derived from existing literature** because it has to be **tailored for the specific model/framework**. In other words, we propose a new graph generative model and theoretically prove it also enjoys nice permutation equivariant/invariant properties.
>
> We will revise this paper's introduction to highlight our unique contributions better.
>
> ---
>
> > **Q2.** no implementation available
>
> **A2.** We have provided our code as a ZIP file in the **supplementary material on OpenReivew**. We are sorry that we forgot to mention this in the paper. We will also open-source our code once this paper is accepted.
>
> ---
>
> > **Q3.** there is a disconnection between the proposed motivation (sampling motivation) and the (theoretical) discussion of the proposed method. The motivation for making the original discrete transition process continuous is to improve the sampling efficiency. Has this been achieved in the proposed method? If so, what is the improvement and where exactly does this improvement come from?
>
> **A3.** Thanks for raising this question to improve the presentation of this paper. We clarify your concerns from the following three perspectives:
>
> 1. Yes, **our proposed method successfully improves the generation efficiency**. The detailed comparison is in Table 9, which shows that the generation quality is still very good even after decreasing the sampling steps from 500 to 100 (i.e., only with 1/5 wall-clock sampling time).
>
> 2. The reason why the proposed model can achieve such a quality-efficiency tradeoff is that **the step size of approximate simulating methods (e.g., $\tau$-leaping [1]) for CTMC determines the accuracy vs. efficiency trade-off**: smaller step sizes generally lead to more accurate simulations but require more computational time and larger step sizes can be faster but with lower simulating accuracy.
>
> 3. To achieve such sampling flexibility, this paper presents a CTMC-based graph diffusion generative framework. We agree with the reviewer that our theoretic analysis is not related to the generation efficiency; however, **analysis in Section 3.6 illustrates that our CTMC-based framework is suitable for graph generation** (i.e., with important permutation equivariant/invariant properties) and **Theorem 3.3 motivates the loss function**, which is critical for the model training.
>
> We will highlight the connection between the motivation and the proposed method in revised versions.
>
> ```
> Reference:
>
> [1] Campbell, Andrew, et al. "A continuous time framework for discrete denoising models." NeurIPS 2022.
>
> [2] Vignac, Clement, et al. "DiGress: Discrete Denoising diffusion for graph generation." ICLR 2023.
>
> [3] Austin, Jacob, et al. "Structured denoising diffusion models in discrete state-spaces." NeurIPS 2021.
>
> ```

---

> > ### Comment · Reviewer_XoYX · 2024-08-10
> >
> > Thank you for the diligent response. However, I still found my main concerns not fully addressed.
> >
> > As the main motivation for using a continuous Markov model instead of a discrete one is to improve the sampling efficiency, one should expect to see more evaluation and evidence to support and validate that this has been achieved.
> >
> > In addition, the result in Table 9 shows that the proposed method is robust to the sampling step and as mentioned in the paper, DIGRESS also has this property. This does not show there is an improvement in sampling efficiency.
> >
> > Q: Why can't $\tau$-leaping be used in the discrete model? Can it be applied in other continuous graph generation such as GDSS?

---

> > > ### Author Response · Authors · 2024-08-11
> > > **Further Discussion**
> > >
> > > Dear Reviewer XoYX,
> > >
> > > We appreciate your insightful response and are glad to clarify your concerns. We sincerely invite the reviewer to have further discussions.
> > >
> > > > **Q1.** Why can't $\tau$-leaping be used in the discrete model? Can it be applied in other continuous graph generation such as GDSS?
> > >
> > > **A1.** We clarify this as follows.
> > >
> > > 1. **$\tau$-leaping [1-3] is designed for continuous-time discrete-state** stochastic processes (CTMC). It **cannot** be applied to discrete-time models (e.g., Digress [4]) because $\tau$-leaping needs the rate matrices as input, but **discrete-time models do not have rate matrices**.
> > >
> > > 2. It **cannot** be applied to GDSS because GDSS is a **continuous-state** model, which is not characterized by the rate matrices.
> > >
> > > > **Q2.** The result in Table 9 shows that the proposed method is robust to the sampling step and as mentioned in the paper, DIGRESS also has this property. This does not show there is an improvement in sampling efficiency.
> > >
> > > **A2.** We would like to answer the reviewer' question from the following perspectives:
> > >
> > > 1. Theoretically, Digress cannot have such sampling flexibility. For example, **if the model is trained with 500 forward steps, technically, its reverse process has to include 500 steps**. In other words, for a Digress model trained with 500 forward steps, doing the reverse process in 100 steps is hard to ensure that the generated samples are from the desired generation distribution.
> > >
> > > 2. Empirically, we have conducted the following experiments to answer the reviewer's question better. We train a Digress model in **500 forward steps** and use it to **do reverse sampling in 500 steps**. We artificially freeze the reverse transition probability in every $\mathrm{round}(500/k)$ steps, where $k$ is the number of inferences. In this way, we have a fair comparison to show the sampling robustness regarding the number of model inferences in Table 1. The metric is validity, and the results of DisCo are shown in Table 4 of our submitted paper. Both DisCo and Digress use the graph Transformer as the backbone and the marginal distribution as the reference distribution. **DisCo is consistently more robust regarding the decrease in the number of inferences. In other words, DisCo has a better quality-efficiency tradeoff.**
> > >
> > > | \# of inferences | DisCo | Digress |
> > > |---|---|---|
> > > | 500 | 99.3$\pm$0.6 | 98.9$\pm$0.4 |
> > > | 100 | 98.7$\pm$0.5 | 95.0$\pm$1.1 |
> > > | 30 | 97.9$\pm$1.2 | 92.1$\pm$0.7 |
> > > | 10 | 95.3$\pm$1.9 | 87.9$\pm$0.8 |
> > > | 5 | 93.0$\pm$1.7 | 80.1$\pm$1.4 |
> > > | 1 | 76.1$\pm$2.3 | 67.1$\pm$2.9 |
> > > Table 1. Quality-efficiency tradeoff comparison on QM9
> > >
> > > 3. In addition, we did not find results from the Digress paper [4] showing that Digress is robust to the step size. If the reviewer could kindly give us a point on related results/statements, we will be happy to check them out.
> > >
> > > > **Q3.** As the main motivation for using a continuous Markov model instead of a discrete one is to improve the sampling efficiency, one should expect to see more evaluation and evidence to support and validate that this has been achieved.
> > >
> > > **A3.** As we mentioned in the submitted paper, Tables 4 and 9 show that the model's performance is robust when the sampling steps are decreased, achieving a decent generation quality-efficiency tradeoff. Lines 844-846 in the appendix detail the sampling steps used to search for the best results, much smaller than Digress's number of sampling steps. In addition, the results from Table 1 in this response should be supplementary to show the generation efficiency of DisCo, which will be included in the revised paper.
> > >
> > > ```
> > > Reference:
> > >
> > > [1] Campbell, Andrew, et al. "A continuous time framework for discrete denoising models." NeurIPS 2022.
> > >
> > > [2] Gillespie, Daniel T. "Approximate accelerated stochastic simulation of chemically reacting systems." The Journal of chemical physics 115.4 (2001): 1716-1733.
> > >
> > > [3] Wilkinson, Darren J. Stochastic modelling for systems biology. Chapman and Hall/CRC, 2018
> > >
> > > [4] Vignac, Clement, et al. "DiGress: Discrete Denoising diffusion for graph generation." ICLR 2023.
> > > ```

---

> > > > ### Comment · Reviewer_XoYX · 2024-08-14
> > > >
> > > > I thank the authors for their diligent responses. Most of my concerns and questions are addressed and I have raised my score accordingly.
> > > > I will suggest the authors to make further emphasis on the sampling aspect in both the technical and experiment section for revised version, as that is the main the motivation of the paper.

---

> > > > > ### Author Response · Authors · 2024-08-14
> > > > >
> > > > > We sincerely thank the reviewer for insightful questions, suggestions, and discussion. We appreciate your recognition of our work and will revise the paper according to the valuable feedback provided!

---

### Official Review · Reviewer_ohXc · 2024-07-09

**Soundness:** 3
**Presentation:** 2
**Contribution:** 2
**Rating:** 5
**Confidence:** 3

**Summary:**

This paper categorizes existing graph diffusion models into 4 types according to whether the space of states and time steps are discrete or continuous.  The authors design DISCO, the first discrete-state continuous-time graph diffusion generative model. By incorporating continuous-time Markov chains while preserving the discrete nature of graph data, DISCO enables more flexible sampling and decouples it from model’s training. Furthermore, this paper explores a novel and concise message-passing neural network as a replacement for complex graph transformer, while maintaining competitive performance.

**Strengths:**

1.	The motivation of this work is clear and easy to understand.
2.	This paper provides comprehensive mathematical proofs to help to clearly understand how to apply continuous-time Markov chains to graph diffusion generative models

**Weaknesses:**

1.	Some symbols in the formulas of this paper are not clear in the context, which reduces readability
2.	This paper seems to merely combine continuous-time Markov chains with discrete graph data, essentially integrating two previous works, and the innovation is slightly insufficient.

**Questions:**

Besides more flexible sampling,  do graph diffusion models based on continuous time have any other advantages compared to discrete-time models? Does the flexibility of sampling provide any particular benefits to generating new data?

**Limitations:**

Yes, the authors addressed limitations, potential negative societal impact, and mitigation.

---

> ### Author Rebuttal · Authors · 2024-08-05
>
> Dear reviewer ohXc, we thank you for your tremendous time and effort in reviewing our paper, and we are glad that you recognize this paper's clear motivation and theoretical contribution. For your questions, our point-to-point responses are as follows.
>
> > **Q1.** Some symbols in the formulas of this paper are not clear in the context, which reduces readability
>
> **A1.** We are sorry that this paper's readability is not satisfactory, and we will try our best to improve its clarity. If the reviewer could kindly point out  specific notations that are not well-explained in the context, it will help us **fix them in the revised versions**.
>
> ---
>
> > **Q2.** This paper seems to merely combine continuous-time Markov chains with discrete graph data, essentially integrating two previous works, and the innovation is slightly insufficient.
>
> **A2.** Although there have been prior works (e.g., [1]) on CTMC-based diffusion models, we make the following new contributions:
>
> 1. We derive a CE-based loss function that differs from the ELBO-based loss in existing CTMC-based diffusion [1]. The CE-based loss has been widely used in discrete diffusion [2-3] but still lacks a theoretical foundation. To bridge this gap, we prove our Theorem 3.3, showing that **the CE-based loss is an upper bound of the estimation error of reverse rate matrices**.
>
> 2. Existing CTMC-based diffusion models are not permutation-equivariant, thus unsuitable for graph generation. We propose **a permutation-equivariant version of the CTMC formulation** to address this issue. We remark that it is a highly nontrivial modification of existing discrete-time graph diffusion models. This is because discrete-time diffusion has a simpler reverse process, but the reverse process of CTMC is different and much more complicated.
>
> We will revise this paper's introduction to highlight our unique contributions better.
>
> ---
>
> > **Q3.** Besides more flexible sampling, do graph diffusion models based on continuous time have any other advantages compared to discrete-time models? Does the flexibility of sampling provide any particular benefits to generating new data?
>
> **A3.** Theoretically, CTMC-based graph diffusion generative modeling should achieve comparable performance with other one-shot diffusion frameworks (e.g., Digress [2]) if the GNN has sufficient expressiveness. The main advantage of our method, and also the focus of this paper, is to provide a flexible trade-off between sample quality and generation efficiency. For example, Table 9 in the Appendix shows that **even after decreasing the sampling steps from 500 to 100 (only with 1/5 sampling time), our method can still achieve good generation quality**. In addition, continuous-time models (including our proposed model) have greater potential, compared to the discrete-time models, to **inspire future efficient sampling methods**, e.g., [4] is inspired from [5], a continuous-time model.
>
>
> ```
> Reference:
>
> [1] Campbell, Andrew, et al. "A continuous time framework for discrete denoising models." NeurIPS 2022.
>
> [2] Vignac, Clement, et al. "DiGress: Discrete Denoising diffusion for graph generation." ICLR 2023.
>
> [3] Austin, Jacob, et al. "Structured denoising diffusion models in discrete state-spaces." NeurIPS 2021.
>
> [4] Zhang, Qinsheng, and Yongxin Chen. "Fast Sampling of Diffusion Models with Exponential Integrator." ICLR 2023.
>
> [5] Song, Yang, et al. "Score-Based Generative Modeling through Stochastic Differential Equations." ICLR 2021.
>
> ```

---

### Official Review · Reviewer_Tpg4 · 2024-07-12

**Soundness:** 3
**Presentation:** 3
**Contribution:** 3
**Rating:** 6
**Confidence:** 3

**Summary:**

The paper introduces a novel graph generation method based on the continuous-time Monte Carlo Markov Chain by adapting the previous work [7]. The proposed architecture comes in two variants: one utilizing a graph Transformer and another using a regular message-passing neural network. The performance of the model is evaluated on artificial planar and SBM networks, as well as on real datasets for the molecule graph generation task.

**Strengths:**

- The paper proposes a novel graph generation method relying on a continuous-time Markov Chain-based diffusion generative model.
- The proposed architecture is supported by theoretical analysis.
- The model is evaluated on both artificial and real datasets and shows comparable performance with respect to the state-of-the-art methods.

**Weaknesses:**

- In the real datasets, for the molecule graph generation task, baseline methods such as GraphINVENT and DiGress show comparable or better performance in some metrics.
- The paper relies on adapting a previous continuous-time Markov Chain-based diffusion generative model for the graph domain. Thus, it doesn’t present substantial new material in terms of novelty.

**Questions:**

**Questions**
- The same metrics are not used across various tables (Tables 2 and 6-8), and some tables lack error terms. This inconsistency makes it difficult to compare results comprehensively. Can the authors update the results to enhance the clarity and reliability of the results?
- Can the proposed architecture be applicable to generate weighted and directed networks?
- The related work section is very brief, possibly due to page limitations but an extended version can be added in the appendix.

**Typos and Errors**
- In Lines 94-95, the set membership symbol should be the subset symbol.
- In Lines 193-197, it is assumed that J_{f_i, f^bar_i} follows Poisson distribution so how can the conditions \sum_{ f^bar_i} J_{f_i, f^bar_i} = 1 and J_{f_i, s} > 1 hold for some s because all entries are non-negative?

**Limitations:**

The authors addressed the limitations of the model in the appendix.

---

> ### Author Rebuttal · Authors · 2024-08-05
>
> Dear reviewer Tpg4, we appreciate your tremendous time and effort in reviewing our paper and are glad that you recognize our proposed model, theoretical contribution, and experiments. For your questions, our point-to-point responses are as follows.
>
> ---
>
> > **Q1.** Baseline methods, e.g., GraphINVENT and DiGress, show comparable or better performance in some metrics and datasets.
>
> **A1.** Theoretically, **our CTMC-based model with sufficiently expressive GNNs should perform comparably to other one-shot diffusion models (e.g., Digress)**. The expressiveness comparison between autoregressive models (e.g., GraphINVENT) and one-shot diffusion models is difficult. We would like to emphasize that the main advantage and focus of our method is a **flexible sample quality vs. generation efficiency trade-off**. E.g., Table 9 in the Appendix shows our method's good generation quality even after decreasing the sampling steps from 500 to 100.
>
> ---
>
> > **Q2.** The paper adapts a previous CTMC-based diffusion model for the graph domain. It doesn’t present substantial novel material.
>
>
> **A2.** Although there have been prior works (e.g., [1]) on CTMC-based diffusion models, we make the following new contributions:
>
> 1. We derive a CE-based loss that differs from the existing ELBO-based loss [1]. The CE-based loss has been widely used in discrete diffusion [2-3] but still lacks a theoretical foundation. To bridge this gap, our Theorem 3.3 shows that **the CE-based loss is an upper bound of the estimation error of reverse rate matrices**.
>
> 2. Existing CTMC-based models are not permutation-equivariant, thus unsuitable for graph generation. We propose **a permutation-equivariant CTMC formulation** to address this issue. Remark that it is a highly nontrivial modification of existing discrete-time graph diffusion models. This is because discrete-time diffusion has a simpler reverse process, but the reverse process of CTMC is much more complicated.
>
> We will revise this paper's introduction to highlight our unique contributions better.
>
> ---
>
> > **Q3.** Metrics are different in Tables 2 and 6-8, and some tables lack error terms.
>
> **A3.** We appreciate this suggestion and are eager to enhance this paper's clarity. We clarify your concern regarding metrics and error terms:
>
> 1. To make a fair comparison, we **keep the metrics consistent with Digress [2]**, a published SOTA solution.
>
> 2. Some of the metrics are **specific to datasets** (e.g., MOSES [4] and GuacaMol [5]). It is hard to ensure all the datasets have the same metrics. We invite the reviewer to check Sections F.2.2 and F.2.3 in the Appendix, which detail these metrics.
>
> 3. Error terms for MOSES and GuacaMol are not included because their computations are too expensive for multiple runs (lines 825-826 in the Appendix).
>
> ---
>
> > **Q4.** Can the proposed architecture be applicable to weighted and directed networks?
>
> **A4. Yes, our method applies to both weighted and directed graphs. For weighted graphs, the edge weights need to be discretized**. E.g., for edge weights in $[0,2]$, it could be discretized in $\\{0.0,0.1,\dots,1.9,2.0\\}$ and viewed as discrete edge states. Then this paper's model can be used. Note that discretization has been widely used to generate images [6]. **Our model can also be applied to directed graphs**. In our implementation, for undirected graphs, only the upper-triangle of the edge type matrix $\mathbf E$ will be sampled and the matrix will be symmetrized. **If the dataset is directed, every entry of the matrix $\mathbf E$ will be sampled.**
>
> ---
>
> > **Q5.** An extended related work section can be added.
>
> **A5.** We appreciate this great suggestion and will provide a more comprehensive related work section in the revised version.
>
> ---
>
> > **Q6.** In Lines 94-95, the set membership symbol should be the subset symbol.
>
> **A6.** Sorry for the confusion; we clarify it as follows. For $\mathbf{E} = \\{e^{(i,j)}\\}\_{i,j\in\mathbb{N}^{+}\_{\leq n}}\in\\{1,\dots, a+1\\}^{n\times n}$, $\\{1,\dots, a+1\\}^{n\times n}$ denotes **the set of $n\times n$ matries** and each matrix's entries are in $\\{1,\dots, a+1\\}$ (by the definition of Cartesian product); $\mathbf E$ should be an $n\times n$ matrix so the $\in$ symbol is correct. However, we acknowledge that **the notation $\mathbf E = \\{e^{(i,j)}\\}\_{i,j\in\mathbb{N}^{+}\_{\leq n}}$ is not standard. We will change to a more standard $(\cdot)$-based notation** (e.g. from https://en.wikipedia.org/wiki/Matrix_(mathematics)) in the revised version and our final notation is $\mathbf{E} = (e^{(i,j)})\_{i,j\in\mathbb{N}^{+}\_{\leq n}}\in\\{1,\dots, a+1\\}^{n\times n}$. The notation of the vector $\mathbf F$ will be changed similarly.
>
> ---
>
> > **Q7.** How can the conditions  $\sum\_{\bar{f^i}}J\_{f^i,\bar{f^i}} = 1$ and $J\_{f^i,s}>1$ hold for some $s$ because all entries are non-negative?
>
> **A7.** We thank the reviewer for catching this important typo. The reviewer is right. Every $J\_{f^i,\bar{f^i}}$ follows Poisson distribution (line 193 in our paper) whose possible values are non-negative. **Line 196 should be "if $\sum\_{\bar{f^i}}J\_{f^i,\bar{f^i}} = 1$ and $J\_{f^i,s}=1$, ..."**
>
>
> ```
> Reference:
>
> [1] Campbell, Andrew, et al. "A continuous time framework for discrete denoising models." NeurIPS 2022.
>
> [2] Vignac, Clement, et al. "DiGress: Discrete Denoising diffusion for graph generation." ICLR 2023.
>
> [3] Austin, Jacob, et al. "Structured denoising diffusion models in discrete state-spaces." NeurIPS 2021.
>
> [4] Polykovskiy, Daniil, et al. "Molecular sets (MOSES): a benchmarking platform for molecular generation models." Frontiers in pharmacology 2020.
>
> [5] Brown, Nathan, et al. "GuacaMol: benchmarking models for de novo molecular design." Journal of chemical information and modeling 2019.
>
> [6] Razavi, Ali, et al. "Generating diverse high-fidelity images with vq-vae-2." NeurIPS 2019.
> ```

---

> > ### Comment · Reviewer_Tpg4 · 2024-08-09
> > **Thanks for your response**
> >
> > I appreciate the authors' efforts in addressing the points I’ve indicated, and I increased my score.

---

> ### Author Response · Authors · 2024-08-11
>
> We sincerely appreciate your thoughtful review and are glad to have the opportunity to clarify your questions. Thank you for your recognition of our work!

---

### Author Rebuttal · Authors · 2024-08-05

Dear reviewers, we thank you for your time and effort in providing valuable reviews for our paper. We appreciate the reviewers' recognition that our theoretic results support the proposed method (Tpg4 and ohXc), that our experiments are comprehensive (Tpg4), that our paper is clear and easy to understand (XoYX and 9BTR), that the method is technically sound (XoYX). We also thank the reviewers for their constructive comments to improve this paper further. Our responses to your questions are provided below. We sincerely invite reviewers for further discussion.

---

### Decision · Program_Chairs · 2024-09-25

**Decision:**

Accept (poster)

**Comment:**

The paper proposes a discrete-state continuous-time diffusion model for graph generation. The main novelty of the model stems from its ability to directly deal with the discrete nature of graphs in the generation process while simultaneously being flexible by relying on a continuous-time sampling mechanism. These aspects have been highlighted by the reviewers -- making a solid contribution to the field. The response of the authors to the different questions raised further clarified several aspects of the paper. Therefore, I recommend accepting the paper.